

# OBELICS: An Open Web-Scale Filtered Dataset of Interleaved Image-Text Documents

**Hugo Laurençon**[*,1,2]  **Lucile Saulnier**[*,1]  **Léo Tronchon**[*,1]
**Stas Bekman**[*,1]  **Amanpreet Singh**[*,1]  **Anton Lozhkov**[1]
**Thomas Wang**[1]  **Siddharth Karamcheti**[1,3]  **Alexander M. Rush**[†,1]
**Douwe Kiela**[†,1,3]  **Matthieu Cord**[†,2]  **Victor Sanh**[*,†,1]

[*]Equal contributions, [†]Senior contributions

hugo@huggingface.co

[1]Hugging Face [2]Sorbonne Université [3]Stanford University

## Abstract

Large multimodal models trained on natural documents, which interleave images and text, outperform models trained on image-text pairs on various multimodal benchmarks. However, the datasets used to train these models have not been released, and the collection process has not been fully specified. We introduce the OBELICS dataset, an open web-scale filtered dataset of interleaved image-text documents comprising 141 million web pages extracted from Common Crawl, 353 million associated images, and 115 billion text tokens. We describe the dataset creation process, present comprehensive filtering rules, and provide an analysis of the dataset's content. To show the viability of OBELICS, we train vision and language models of 9 and 80 billion parameters named IDEFICS, and obtain competitive performance on different multimodal benchmarks. We release our dataset, models and code.[1]

## 1 Introduction

Recent systems demonstrate the effectiveness of training large multimodal models such as Flamingo on naturally occurring multimodal documents (Alayrac et al., 2022; Aghajanyan et al., 2022; Huang et al., 2023). A multimodal document is a succession of text paragraphs interleaved by images, such as web pages that contain images. Models trained on these web documents outperform vision and language models trained solely on image-text pairs on various benchmarks (Alayrac et al., 2022). They can also generate long and coherent text about a set of multiple images.

While these results are compelling, they have not been replicable. The datasets used in these works are not publicly available, and relatively little information is known about their creation process and composition. This state motivates the creation of large-scale collections of high-quality multimodal web documents to support the creation of the next generation of models.

We take inspiration from existing large open image-text datasets such as LAION (Schuhmann et al., 2022) and COYO (Byeon et al., 2022), comprised of hundreds of millions of image-text

---

OBELICS: https://huggingface.co/datasets/HuggingFaceM4/OBELICS
[1]OBELICS reproduction code: https://github.com/huggingface/OBELICS
IDEFICS models: https://huggingface.co/HuggingFaceM4/idefics-80b

37th Conference on Neural Information Processing Systems (NeurIPS 2023) Track on Datasets and Benchmarks.

Image-Text Pairs

Multimodal Document

Tottenham vs Chelsea Live Streaming

Tottenham Spurs vs Chelsea Live Streaming

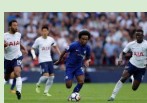

The match between Tottenham Spurs vs Chelsea will kick off from 16:30 at Tottenham Hotspur Stadium, London.

The derby had been played 54 times and the Blues have dominated the Spurs. Out of 54 matches played, Chelsea has won 28 times and Spurs had only won 7 times. The remaining 19 matches had ended in draw.

However, in recent 5 meetings, Spurs had won 3 times where Chelsea had won the other two times. …

Figure 1: A comparison of extraction from the same web document. For image-text pairs, the alt-text of images is often short or non-grammatical. For OBELICS, the extracted multimodal web document interleaves long-form text with the images on the page.

pairs obtained through web crawling. These datasets have been critical to developing and replicating numerous recent multimodal models (Radford et al., 2021; Wang et al., 2022; Yu et al., 2022; Wang et al., 2022; Liu et al., 2023). While this approach allows for building extremely large and diverse training datasets, we note several limitations to using only image-text pairs. From a language perspective, these datasets rely primarily on alt-text, meaning the text given is brief, captures an approximate snapshot of the image's content, and often lacks grammatical correctness. From a document perspective, image-text pairs remove an image from its natural context on a page and its relationship with other documents.

In this work, we introduce OBELICS[2], an openly-accessible curated web-scale dataset consisting of 141 million multimodal English web documents which contain 353 million associated images and 115 billion tokens. OBELICS collects full multimodal documents interleaving text and images as shown in Figure 1. We describe the dataset creation process, outline the filtering and curation steps and shed light on the dataset's content and limitations. To demonstrate the viability of OBELICS, we train IDEFICS, an 80 billion parameter multimodal model and show competitive performance against large-scale multimodal models such as Flamingo (Alayrac et al., 2022).

## 2  Related Works

**Image-text pairs datasets**   The largest multimodal datasets, such as LAION (Schuhmann et al., 2021, 2022), Conceptual Captions (Sharma et al., 2018; Changpinyo et al., 2021), ALIGN (Jia et al., 2021), COYO (Byeon et al., 2022), and DataComp (Gadre et al., 2023), contain billions of image-text pairs and are usually obtained through web-crawling and alt-text extraction. A variety of multimodal models have been trained on this type of dataset: multimodal encoder models which use a contrastive objective (Radford et al., 2021; Wang et al., 2022), image generation based on Transformers or diffusion processes (Nichol et al., 2022; Ramesh et al., 2022; Rombach et al., 2021; Saharia et al., 2022). While the scale of these datasets makes them attractive candidates for training, our work focuses on extracting images and the textual context in which they appear instead of extracting the associated alternative text.

**Web document datasets**   Insights from scaling language models (Kaplan et al., 2020; Hoffmann et al., 2022) emphasize the need for increasingly bigger datasets. For instance,

---

[2]Open Bimodal Examples from Large fIltered Commoncrawl Snapshots

LLaMA (Touvron et al., 2023) was trained on a dataset of 1.4T tokens created exclusively from openly accessible English web content. The authors noticed that an even bigger dataset would have benefited the model. To address that need, multiple web-scale datasets have been introduced and made available: c4 (Raffel et al., 2019), ROOTS (Laurençon et al., 2022), Pile (Gao et al., 2020), OSCAR (Ortiz Suárez et al., 2020). Although OBELICS falls in the same category of making accessible large collections of curated web documents, the additional extraction of images changes the nature of the resulting dataset. It allows training models with additional vision capabilities.

**Multimodal web document datasets** The recent most performant vision and language models are trained on large sets of multimodal web documents. For instance, Flamingo (Alayrac et al., 2022), an 80 billion multimodal model, was trained on a mix of 2.1 billion image-text pairs, 27 million video-text pairs, and 43 million multimodal web documents. The latter, called M3W, includes 185 million images. Similarly, KOSMOS-1 (Huang et al., 2023) was trained on a mixture containing 71 million multimodal web documents. However, in both cases, the dataset is not publicly available, and little information is accessible as to the dataset's content, the strategies employed to create that dataset (including filtering strategies), and the quality of the resulting web documents, which ultimately hinders further research.

Concurrently to our work, the Multimodal C4 (mmc4) dataset (Zhu et al., 2023) was recently made accessible. It consists of 103 million multimodal web documents that include 585 million images. Although there are similarities between our datasets, it is important to highlight particular distinctions. First, our dataset is based on more recent documents from February 2020 to February 2023, whereas mmc4 uses documents from April 2019. Additionally, our filtering heuristics appear to be more comprehensive: we leverage the HTML DOM trees to filter out undesirable texts and images, whereas mmc4 uses the HTML to find images in order to merge them with the original C4 dataset by solving a bipartite assignment problem based on CLIP model similarities. Last, we implement additional deduplication steps at the image, document, and paragraph levels.

## 3   Creation of the Multimodal Web Document Dataset

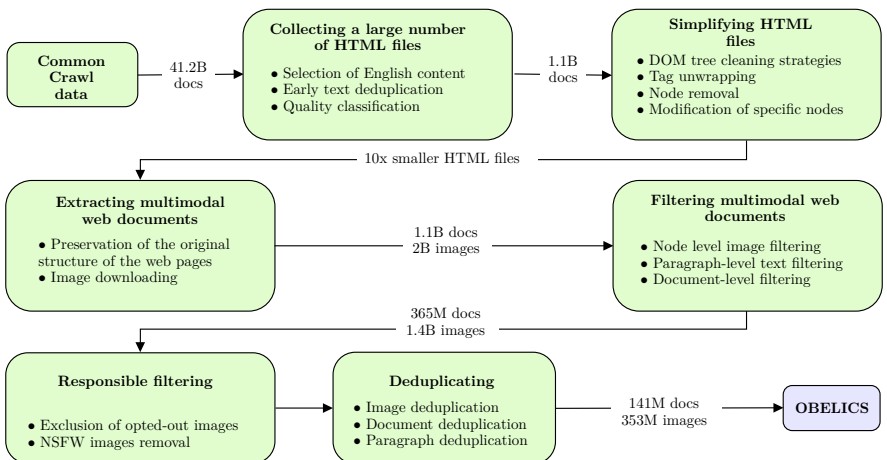

Figure 2: Overview of the steps involved in creating OBELICS.

This section provides an overview of the critical choices of the creation and filtering process. Figure 2 gives a high-level summary of the main steps involved. Many details are omitted from this section, and we invite the reader to refer to the appendix A.1 for completeness.

## 3.1 Collecting a Large Number of HTML Files

First, we collect a vast amount of raw web documents by considering the 25 most recent Common Crawl dumps at the time of the creation, spanning from February 2020 to January/February 2023[3]. We extract the main text from the documents while discarding documents with text of insufficient quality. This process results in 41.2 billion documents.

To filter out non-English content, we apply the FastText classifier (Joulin et al., 2017) to the extracted text, which removes 63.6% of the documents. We perform a MinHash (Broder, 1997) deduplication to remove duplicate content. Additionally, we filter out documents with significant proportions of repeated paragraphs and n-grams, following the methodology used in MassiveText (Rae et al., 2022). Previous studies (Lee et al., 2022; Abbas et al., 2023) have demonstrated the prevalence of duplication in crawled data and the benefits of training on deduplicated data.

Similar to Brown et al. (2020), we employ a logistic regression classifier with hashed token frequencies to ensure high-quality text. This classifier, trained using curated datasets like Wikipedia or OpenWebText (Gokaslan and Cohen, 2019) as positive examples and documents sampled from Common Crawl as negative ones, is fast and effective at detecting human-written text. After these steps, we are left with 1.1 billion documents and their HTML sources from the associated Common Crawl WARC files.

## 3.2 Simplifying HTML Files

The original HTML content of a document contains a wealth of valuable information that proves highly beneficial in the process of filtering out undesirable text and images. Therefore, we prioritize pre-processing the raw HTML into simplified HTML, making the subsequent extraction of textual and visual elements more efficient.

To this aim, we devise multiple pre-processing strategies for an HTML DOM tree. By manually inspecting instances of all HTML nodes, we differentiate nodes likely to contain relevant texts or images from those that should be discarded, and we formulate specific rules for each type of node. After these pre-processing steps, the resulting simplified HTML files are more than ten times smaller and have been stripped of a large proportion of generic text (spam, ads, boilerplate template, etc.) and generic images, such as logos, while retaining the relevant content.

## 3.3 Extracting Multimodal Web Documents

In this step, we transform the simplified HTML files previously obtained into a structured web multimodal web document format. This format consists of interleaved texts and images.

We meticulously preserve the original structure of the web pages from the simplified HTML files by extracting the texts and image links while maintaining their rendering defined by the DOM tree. Given that each HTML tag denotes a distinct separation between the preceding and subsequent nodes, we leverage that information to retain line breaks and line feeds on the original page, preserving the formatting and visual rendering of the content.

We obtain 3.6 billion image links and successfully download 55% of them (approximately 2 billion images).

## 3.4 Filtering Multimodal Web Documents

The filtering process comprises two distinct steps operating at different granularity levels. In the first step, filtering occurs at the node level for images and the paragraph level for text. This step guarantees that only high-quality and relevant images and paragraphs are retained. Each paragraph or image is evaluated based on specific criteria and may undergo modifications or be eliminated if necessary. The second step, conducted at the document level, involves deciding whether to retain or discard the output documents obtained from the

---

[3] https://commoncrawl.org/

first step. Most text filters used in both steps are primarily derived from Laurençon et al. (2022).

**Node-level image filtering**   We discard images that are too small, excessively large or have disproportionate dimensions. We observe that these images are often indicative of low-quality or irrelevant content. To eliminate some logos and generic images, we remove images whose URLs contain one of the banned sub-strings, like *logo*.

**Paragraph-level text filtering**   We apply multiple filters to text paragraphs to remove undesirable content. Specifically, paragraphs that contain an insufficient number of words are discarded. Additionally, we filter out paragraphs with high repetition ratios, excessive ratios of special characters, low ratios of stop words, low punctuation ratios, high proportions of flagged words associated with adult or inappropriate content, or excessively high perplexity scores (as measured by an n-gram language model trained on Wikipedia (Heafield, 2011)). To identify boilerplate sentences or invitations to share articles on social networks, we create a list of frequently used words associated with these paragraphs and remove paragraphs containing an excessive proportion of words from this list. To further identify machine-generated content, we extract words from web-crawled documents to form a list of common words and discard documents with a low ratio of common words.

**Document-level filtering**   At the document level, we remove all documents with no or excessively high number of images. For text filters, the same filters used at the paragraph level are applied, with sometimes stricter cutoff values.

After these filtering steps, we are left with 365 million web documents and 1.4 billion images. At this step, images can be duplicated across documents.

## 3.5   Responsible Filtering and Deduplication

We take measures to minimize the amount of inappropriate content in the dataset. In particular, based on manual inspections and tool availability, we implement filters to respect data consent and remove images with pornographic content. Additionally, we also heavily deduplicate content.

**Exclusion of opted-out images**   To respect the preferences of content creators, we remove all images for which creators explicitly opted out of AI model training. We used the Spawning API[4] to verify that the images in the dataset respect the original copyright owners' choices.

**Image deduplication based on URL**   Some images could be present across different documents. We observe that it is particularly true for browser-specific icons or common advertisements encountered during the crawling process. To address this issue, we remove all images that appear more than ten times across the entire dataset. We intentionally do not perform strict deduplication, as we notice that when an image is duplicated only a few times across different documents, the surrounding text and contextual information tend to be different. We also deduplicate images within the same document.

**NSFW image filtering**   To reduce explicit adult content, we use an open-source NSFW classifier to remove entire documents containing pornographically classified images. We also filter out images with URLs containing banned sub-strings.

**Document deduplication based on URL and set of images**   We complete the initial deduplication step by forming clusters of documents with the same URLs and retaining the most recent document within each cluster. We repeat this operation by forming clusters of documents containing identical sets of images.

**Paragraph deduplication across documents of the same domain names**   To remove generic spam phrases commonly found at the end of documents, we perform paragraph-level

---

[4] https://api.spawning.ai/spawning-api

exact deduplication within documents sharing the same domain name, resulting in the elimination of approximately 15% of the text.

Following these filtering and deduplication steps, the final dataset contains 141 million documents and 353 million images, of which 298 million are unique. We observe that using stricter values for the filtering steps yields fewer multimodal documents, although not of higher quality. As such, we invite users who are interested in manipulating a smaller subset of OBELICS to start with a random subset.

## 4 Analysis of OBELICS

Figure 1 provides an example showcasing an original webpage alongside the resulting multimodal web document. Extracting and filtering the multimodal document is non-trivial as it requires carefully removing undesirable information on the left, top, and bottom of the page, such as menus and navigation bars. We provide other examples at https://huggingface.co/spaces/HuggingFaceM4/obelics_visualization and in Figures 7, 8 and 9.

Given the scale of OBELICS, it would be prohibitive to describe its content exhaustively. Instead, we provide high-level statistics and analyses that shed light on the dataset's properties.

### 4.1 General Statistics

| Dataset | Images | % unique images | Docs | Tokens | Open |
|---|---|---|---|---|---|
| KOSMOS-1 | - | - | 71M | - | ✗ |
| M3W | 185M | - | 43M | - | ✗ |
| mmc4-ff | 385M | 60.6% | 79M | 34B | ✓ |
| mmc4 | **585M** | - | 103M | 43B | ✓ |
| OBELICS | 353M | **84.3%** | **141M** | **115B** | ✓ |

Table 1: General statistics of OBELICS and the current largest alternatives.

Figure 3: Distribution of images.

Table 1 compares OBELICS against the largest existing alternatives. mmc4-ff is the mmc4 dataset with fewer faces. Our dataset has the highest number of unique documents and total tokens while containing a huge number of images.

It is worth mentioning that we have fewer images than mmc4 (Zhu et al., 2023). This discrepancy can be attributed to two reasons. First, our analysis reveals that mmc4 contains many duplicated images, with only 60.6% being unique compared to 84.3% for OBELICS. We found that images duplicated multiple times often indicate spam or unrelated generic content. Second, mmc4 does not limit the number of images within a document. As a result, the distribution of images across documents is highly uneven, with a substantial portion of them concentrated in documents with excessive image counts (see Figure 3). The images in these documents are often unrelated to each other and exhibit spam or advertisement content. Moreover, these documents often have little text, making them unsuitable for learning the alignment between text and images (see an example in Figure 10).

Figure 4 shows the joint distribution of a number of tokens and a number of images in OBELICS. Although we limit the number of images in a document to 30, we cut the plot at 6 images for clarity. The documents of OBELICS contain a median number of images of 1 and a median number of tokens of 677.

**Perplexity analysis** To assess the quality of our text in comparison to reference datasets used for training large language models, we leverage an n-gram language model trained on Wikipedia (Heafield, 2011; Laurençon et al., 2022). This allows us to compute perplexity

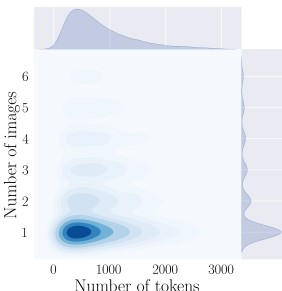

Figure 4: Heatmap displaying the joint distribution of the number of tokens and the number of images in OBELICS documents, accompanied by their respective marginal distributions.

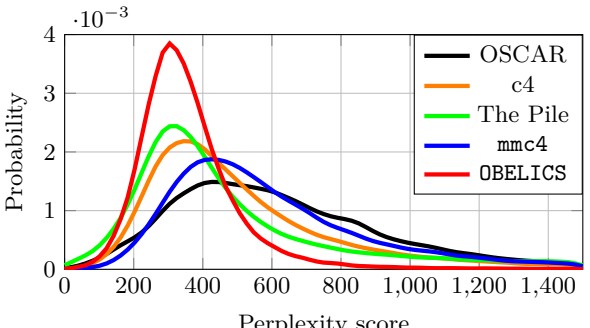

Figure 5: Kernel density estimations representing the distribution of perplexity scores for OBELICS compared to reference datasets. The lower the perplexity for a document, the more it resembles a Wikipedia article.

scores for 100,000 documents from each dataset. Lower perplexity scores indicate a higher resemblance to Wikipedia documents. Figure 5 displays the distributions of these scores. Our results demonstrate that the texts in OBELICS have a significantly lower average perplexity compared to the texts in c4 (Raffel et al., 2019), mmc4 (Zhu et al., 2023), and OSCAR (Ortiz Suárez et al., 2020). Furthermore, our distribution aligns closely with the one from The Pile (Gao et al., 2020), which was thoughtfully curated from diverse, high-quality sources.

### 4.2 Topic Modeling

Similar to Zhu et al. (2023), we employ a Latent Dirichlet Allocation (LDA) (Blei et al., 2003) to understand the diversity of the dataset. The LDA gives us insights into the distribution of topics in the dataset, along with estimated proportions and frequently associated words. Table 5 and 6 present the results of the LDA with respectively 20 and 200 topics, offering both a high-level and a more granular analysis of the dataset's content. We observe that the dataset covers topics ranging from Politics to Health by way of Music. Additionally, we compute the most frequent domains and show that news sites are systematically the most represented (Table 4).

### 4.3 Qualitative Assessment of Dataset Samples

We manually inspect 250 documents from OBELICS to verify the dataset's quality and asses the risks contained in the dataset. We focus on the images' content in relation to the text since it's the core addition compared to a language modeling dataset.

80% of documents have photo images, while 29% have graphic images (drawings, cartoons, etc.). 90% of the documents have all images clearly related to the text content. 30% of documents have images containing at least one written word, and 5% of documents have images that are structured text (slides, tables, scanned documents, etc.), which can help models learn OCR capabilities. 7% of documents have content (images or text) that hasn't been captured by cleaning filters (non-English text, spam or advertisement, etc.). 46% of documents contain images with faces (portraits or group photos). No obvious Personally Identifiable Information (PII) texts were found, except for public personalities and people mentioned in news articles. No NSFW images were found. Only 3% of documents contain images with watermarks, and 2% have images with logos.

## 5 Validating the Viability of OBELICS

To confirm the viability of our dataset, we first show that vision and language models trained on our multimodal web documents outperform the same models trained on image-text pairs on various multimodal benchmarks. Following that, we demonstrate the effectiveness of

`OBELICS` as an alternative to closed datasets by training models of different sizes on par with closed-source models.

**Model details** We follow the Flamingo (Alayrac et al., 2022) architecture closely: we combine two frozen unimodal backbones - LLaMA (Touvron et al., 2023) for the language model, and OpenClip [5] for the vision encoder - add learnable cross-attention Transformer blocks to connect the language and vision blocks. For multimodal web documents, we feed the model sequences corresponding to the succession of text paragraphs and images. For image-text pairs, we form the training sequences by packing images with their captions. The images are encoded with the vision encoder and vision hidden states are pooled with Transformer Perceiver blocks and then fused into the text sequence through the cross-attention blocks. The training objective is the standard next token prediction. For more details, we refer to the original paper.

Following Alayrac et al. (2022), we evaluate our models on a series of multimodal benchmarks spanning visual question answering (VQAv2 (Antol et al., 2015), OKVQA (Marino et al., 2019), TextVQA (Singh et al., 2019), VizWiz (Gurari et al., 2018)), visual dialogs (VisDial (Das et al., 2017)), hateful speech detection (HatefulMeme (Kiela et al., 2020)), image captioning (COCO (Lin et al., 2014), Flickr30k (Young et al., 2014)), and OCR (IIIT5k (Mishra et al., 2012)).

Additional details about the architecture, the training, the compute and the evaluation are present in Appendix A.4.

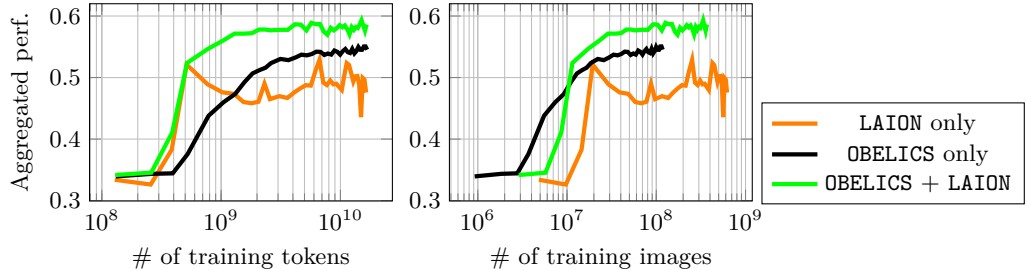

Figure 6: Aggregated 4-shot performance through the training using `LAION` only, `OBELICS` only and a mixture of both. The training sequences from multimodal documents and the packed sequences obtained from image-text pairs have different numbers of images but the same number of tokens. Thus, we plot the performance over two log x-axes. The initial uptick of the model trained on image-text pairs is attributed to the fact the performance on VQA tasks starts by increasing and then slowly degrades.

**Training on different mixture of data** Figure 6 shows the result of the first experiment, which consists in training 9B-parameter models on different mixture of data. Training on multimodal web documents allows reaching the same performance using an order of magnitude fewer images than training on image-text pairs, even though the images from the two datasets come from Common Crawl. This underlines the benefit of having longer text contexts for training multimodal models. Moreover, the model trained on multimodal web documents performs better on average. This is particularly striking on visual question-answering benchmarks on which the model trained on image-text pairs slowly degrades through the training. We note, however, that the model trained on image-text pairs has a slight advantage performance-wise in captioning, classification, and OCR tasks (see more details in Appendix A.4.5). We hypothesize that this is due to the nature of image-text pairs: captions can be seen as fuzzy class labels. Last, similarly to Alayrac et al. (2022), we observe that combining the two types of datasets leads to increased performance for a given number of images, tokens, or training compute.

**Models trained on `OBELICS` achieve competitive performance at different scales** Following these insights, we show that `OBELICS` is a viable open alternative to other datasets.

---

[5] https://laion.ai/blog/large-openclip/

| | Shot | COCO | Flickr30k | VQAv2 | OKVQA | TextVQA | VizWiz | VisDial | HatefulMemes |
|---|---|---|---|---|---|---|---|---|---|
| Flamingo-9B | | 79.4 | 61.5 | 51.8 | 44.7 | 31.8 | 22.8 | 48.0 | 57.0 |
| OpenFlamingo-9B | 0 | 79.5 | 59.5 | 52.7 | 37.8 | 24.2 | 27.5 | - | 51.6 |
| IDEFICS-9B | | 46.0 | 27.3 | 50.9 | 38.4 | 25.9 | 35.5 | 48.7 | 51.8 |
| Flamingo-9B | | 93.1 | 72.6 | 56.3 | 49.3 | **33.6** | 34.9 | 50.4 | 62.7 |
| OpenFlamingo-9B | 4 | 89.0 | 65.8 | 54.8 | 40.1 | 28.2 | 34.1 | - | 54.0 |
| IDEFICS-9B | | 93.0 | 59.7 | 55.4 | 45.4 | 27.6 | 36.9 | 47.9 | 50.7 |
| Flamingo-9B | | 99.0 | **73.4** | 58.0 | 50.0 | **33.6** | 39.4 | 51.2 | 63.9 |
| OpenFlamingo-9B | 8 | 96.3 | 62.9 | 54.8 | 41.1 | 29.1 | 38.5 | - | 54.7 |
| IDEFICS-9B | | 97.0 | 61.9 | 56.4 | 47.7 | 27.5 | 40.4 | 47.6 | 51.1 |
| Flamingo-9B | | 102.2 | 72.7 | 59.4 | 50.8 | 33.5 | 43.0 | **51.3** | **64.5** |
| OpenFlamingo-9B | 16 | 98.8 | 62.8 | 54.3 | 42.7 | 27.3 | 42.5 | - | 53.9 |
| IDEFICS-9B | | 99.7 | 64.5 | 57.0 | 48.4 | 27.9 | 42.6 | - | 50.1 |
| Flamingo-9B | | **106.3** | 72.8 | **60.4** | **51.0** | 32.6 | **44.0** | 50.4 | 63.5 |
| OpenFlamingo-9B | 32 | 99.5 | 61.3 | 53.3 | 42.4 | 23.8 | **44.0** | - | 53.8 |
| IDEFICS-9B | | 98.0 | 64.3 | 57.9 | 49.6 | 28.3 | 43.7 | - | 49.8 |
| Flamingo | 0 | 84.3 | 67.2 | 56.3 | 50.6 | 35.0 | 31.6 | 52.0 | 46.4 |
| IDEFICS | | 91.8 | 53.7 | 60.0 | 45.2 | 30.9 | 36.0 | 48.9 | 60.6 |
| Flamingo | 4 | 103.2 | 75.1 | 63.1 | 57.4 | 36.5 | 39.6 | 55.6 | 68.6 |
| IDEFICS | | 110.3 | 73.7 | 63.6 | 52.4 | 34.4 | 40.4 | 48.4 | 57.8 |
| Flamingo | 8 | 108.8 | 78.2 | 65.6 | 57.5 | 37.3 | 44.8 | 56.4 | **70.0** |
| IDEFICS | | 114.3 | 76.6 | 64.8 | 55.1 | 35.7 | 46.1 | 47.9 | 58.2 |
| Flamingo | 16 | 110.5 | 78.9 | 66.8 | **57.8** | 37.6 | 48.4 | **56.8** | **70.0** |
| IDEFICS | | **116.6** | 80.1 | 65.4 | 56.8 | 36.3 | 48.3 | - | 57.8 |
| Flamingo | 32 | 113.8 | 75.4 | **67.6** | **57.8** | **37.9** | 49.8 | 55.6 | **70.0** |
| IDEFICS | | **116.6** | **81.1** | 65.9 | **57.8** | 36.7 | **50.0** | - | 52.5 |

Table 2: Performance of `IDEFICS` against OpenFlamingo and Flamingo. The evaluations were done with random in-context examples, and in an open-ended setting for VQA tasks. (Task, Metric, Query split): (COCO, CIDEr, test), (Flickr30k, CIDEr, test (Karpathy)), (VQAv2, VQA acc., testdev), (OKVQA, VQA acc., val), (TextVQA, VQA acc., val), (VizWiz, VQA acc., testdev), (VisDial, NDCG, val), (HatefulMemes, ROC-AUC, test seen).

We train `IDEFICS`, an 80 billion parameters Flamingo-like model on a mixture of image-text pairs from LAION (Schuhmann et al., 2022), openly accessible captioning datasets (Singh et al., 2022), `OBELICS` and multimodal web documents obtained from Wikipedia using a similar extraction strategy. We also train a smaller version of 9 billion parameters, `IDEFICS-9B`. We compare these models against OpenFlamingo v2 (Awadalla et al., 2023) and Flamingo of the same sizes and trained on a similar mixture of multimodal web documents and image-text pairs. We report the results in Table 2.

`IDEFICS` is often on par with Flamingo on various multimodal benchmarks. Out of the 8 evaluation tasks, with 32 in-context examples, it either performs better or obtain the same result as Flamingo on 4 of them. At the 9 billion parameter scale, we are still behind Flamingo-9B. However, it is important to highlight that we outperform OpenFlamingo-9B, which was trained on `mmc4`, in terms of aggregated performance. We achieved a score of 56.5, compared to their score of 55.8, by selecting the best performance across all numbers of in-context examples for each task. This highlights the advantages of `OBELICS` as an open alternative to a multimodal web document dataset.

# 6 Conclusion

With the goal of supporting open-source large multimodal models, we introduce `OBELICS`, an open web-scale collection of filtered interleaved multimodal web documents based on Common Crawl snapshots. We document a collection and filtering process that balances the scale and removal of undesirable texts and images while addressing some of the well-documented ethical concerns of large-scale multimodal datasets, notably data consent and pornographic content. To demonstrate the usefulness of models trained on multimodal documents, we train `IDEFICS` on `OBELICS` and show that it is a viable alternative to closed datasets. Open datasets of multimodal documents with scale, quality, and diversity of sources can help support the ability to train competitive open models.

## Acknowledgments and Disclosure of Funding

The authors were granted access to the HPC resources of the Institut du développement et des ressources en informatique scientifique (IDRIS) du Centre national de la recherche scientifique (CNRS) under the allocation 2022-A0121013450 made by Grand équipement national de calcul intensif (GENCI). The initial development of the dataset was done on Jean-Zay cluster of IDRIS, and we thank the IDRIS team for their responsive support throughout the project, in particular Rémi Lacroix. We thank Guillaume Salou for setting up the virtual machines used to download the images of our dataset, and Sebastian Nagel for his valuable assistance in providing insights on Common Crawl. We thank Yacine Jernite and Daniel van Strien for conducting a bias analysis of the models trained on `OBELICS`.

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
