# A  Appendix

## A.1  Creation of the Multimodal Web Document Dataset

### A.1.1  Collecting of a Large Number of HTML Files

Our data collection process begins by considering the 25 most recent Common Crawl[6] dumps available at the time of dataset creation. It contains webpages spanning from February 2020 to January/February 2023. We use a modified version of `readability-lxml`[7] to extract the main text from the pages, discarding any pages that contain text of excessively high perplexity. This process yields a total of 41.2 billion documents.

**Selection of English content**  To identify non-English content, we apply the FastText classifier (Joulin et al., 2017) to the extracted text, effectively filtering out 63.6% of the documents.

**Early text deduplication**  Often, a set of URLs is crawled repeatedly across different Common Crawl snapshots. However, the content of these websites may vary as web administrators make changes over time. Hence, at this stage, we refrain from deduplicating documents based on their URLs. Instead, we perform MinHash (Broder, 1997) deduplication with 16 hashes calculated over 5-grams. To further refine the data, we eliminate documents containing substantial proportions of repeated paragraphs and n-grams, employing the methodology described in MassiveText (Rae et al., 2022). (Lee et al., 2022; Abbas et al., 2023) show that crawled data often contains a significant amount of duplication, and training on deduplicated data can improve performance.

**Quality classification**  We employ a logistic regression classifier with hashed token frequencies to only retain pages containing human-written text, similar to Brown et al. (2020). The classifier is trained using documents from curated datasets, such as Wikipedia and OpenWebText (Gokaslan and Cohen, 2019), as positive examples, and documents sampled from Common Crawl as negative examples. For simplicity, we use a threshold of 0.5 for the probability that a document comes from a curated corpus, which acts as an indicator that a document is human-written.

Following these steps, we obtain 1.1 billion documents and their HTML sources from the associated Common Crawl WARC files.

### A.1.2  Simplifying HTML Files

The original HTML content of a document contains a wealth of valuable information that proves highly beneficial in the process of filtering out undesirable text and images. Therefore, we prioritize pre-processing the raw HTML into simplified HTML, making the subsequent extraction of textual and visual elements more efficient. For this purpose, we use the library `selectolax`[8] that facilitates efficient parsing of HTML files and creates corresponding DOM trees.

**DOM Tree cleaning strategies**  To simplify the DOM trees, we employ several cleaning strategies. Firstly, we convert tags that indicate line breaks (such as `
`) into actual line breaks. Multiple consecutive line breaks and spaces are condensed into a single instance. Additionally, HTML comments are removed from the DOM trees. Furthermore, we implement recursive processes to eliminate empty leaves and unnest nodes. When a parent node lacks attached text and has only one child, the child node replaces the parent node in the DOM hierarchy. We repeat these operations after removing some nodes, and describe this process in the following paragraphs.

---

[6]https://commoncrawl.org/
[7]https://github.com/buriy/python-readability
[8]https://github.com/rushter/selectolax

**Tag unwrapping**   This operation involves removing unnecessary styling applied to displayed text by unwrapping a predefined set of tags given below. By applying this procedure, tags such as `example` are transformed into `example`, eliminating the associated styling elements.

The following tags are unwrapped during the processing of HTML files: `a`, `abbr`, `acronym`, `b`, `bdi`, `bdo`, `big`, `cite`, `code`, `data`, `dfn`, `em`, `font`, `i`, `ins`, `kbd`, `mark`, `q`, `s`, `samp`, `shadow`, `small`, `span`, `strike`, `strong`, `sub`, `sup`, `time`, `tt`, `u`, `var`, `wbr`.

**Node removal**   Following the previous step, we conduct a manual inspection of practical examples encompassing all existing HTML tags. Based on our findings, we establish a curated list that outlines the tags we intend to retain. Any nodes within the HTML DOM tree with tags not included in this list are subsequently removed. We specifically retain tags that define the document structure (e.g., `p` or `h`) and tags associated with media elements (e.g., `img`). However, we opt to remove tags that typically consist of logos, generic content, or spam (e.g., `header`), as well as tags that often contain noisy text related to website navigation (e.g., `li`), or text that poses challenges in terms of linearization (e.g., `table`).

We retain the following tags during the processing of HTML files, as they define the document's structure: `address`, `article`, `aside`, `blink`, `blockquote`, `body`, `br`, `caption`, `center`, `dd`, `dl`, `dt`, `div`, `figcaption`, `h`, `h1`, `h2`, `h3`, `h4`, `h5`, `h6`, `hgroup`, `html`, `legend`, `main`, `marquee`, `ol`, `p`, `section`, `summary`, `title`, `ul`. Additionally, we also preserve the following tags that define media elements: `audio`, `embed`, `figure`, `iframe`, `img`, `object`, `picture`, `video`. Furthermore, we keep the `source` tag as it may contain an interesting attribute.

**Modification of specific nodes**   We then specifically target some `<div>` nodes that contain `footer`, `header`, `navigation`, `nav`, `navbar`, or `menu` as ID or `date` as attribute, as well as CSS rules that possess `footer` or `site-info` as class. These nodes typically contain website navigation content or article dates and are therefore removed. Additionally, we observe that the presence of a CSS rule with the class `more-link` often indicates a distinct shift in topic within the webpage, resembling the start of a new document. To account for this, we replace these nodes with the text `END_OF_DOCUMENT_TOKEN_TO_BE_REPLACED`, which we replace by an end-of-sentence (EOS) token during training.

With these processing steps, we reduce the size of the HTML files by more than 10 on average while preserving the interesting content.

### A.1.3   Extracting Multimodal Web Documents

In this section, we begin with the simplified HTML files obtained from the previous section. Our objective is to transform these files into a structured web document format, which is a sequence of interleaved texts and images.

**Preservation of the original structure of the web pages**   During the extraction process, we meticulously preserve the original structure of the web pages from the simplified HTML files. We extract the texts and image links while maintaining their order of appearance in the DOM tree. Each HTML tag denotes a distinct separation between the preceding and subsequent nodes and we retain any line breaks and line feeds that are present in the original page, preserving the formatting and visual rendering of the content.

**Image downloading**   To download the images, we use the `img2dataset` (Beaumont, 2021) library. We attempt to download a massive collection of 3.6 billion images, of which 55% (approximately 2 billion images) were successfully downloaded. For that, we employ 20 virtual machines. This distributed approach allow us to complete the operation within a few days.

### A.1.4   Filtering Multimodal Web Documents

The filtering process consists of two steps, targeting different levels of granularity. In the first step, filtering occurs at the node level for images and at the paragraph level (separated by line breaks) for text. We evaluate each paragraph or image and we potentially modify or

remove these based on specific criteria. The second step, conducted at the document level, involves deciding whether to retain or discard the output documents from the first step. The majority of the filters for text we use for both steps were adapted from Laurençon et al. (2022).

**Node-level image filtering**   We discard images with formats other than `jpg`, `png` or `webp`, with a side length below 150 pixels or exceeding 20,000 pixels, as well as those with an aspect ratio greater than 2 or less than 1/2. These criteria help exclude images that are too small, excessively large, or have disproportionate dimensions, which are often indicative of low-quality or irrelevant content. To eliminate some logos and generic images, as in (Zhu et al., 2023), we remove images whose URL contains one of the sub-strings *logo*, *button*, *icon*, *plugin* or *widget*.

**Paragraph-level text filtering**   Regarding text paragraphs, we apply a series of filters to remove undesirable or irrelevant content. We discard paragraphs with fewer than 4 words, as they typically contain insufficient information to be considered meaningful. Additionally, we remove paragraphs with a high repetition ratio, indicating potential spam content, and those with an excessive ratio of special characters, often associated with irrelevant or low-quality text.
Furthermore, we filter out paragraphs with a low ratio of stop words, as it is often indicative of machine-generated or nonsensical content. Similarly, we exclude paragraphs with a low punctuation ratio, as they typically indicate poor-quality texts. We also consider the flagged word ratio, removing paragraphs with a high proportion of flagged words associated with adult or inappropriate content. We also use KenLM (Heafield, 2011) models trained on Wikipedia to filter out paragraphs with excessively high perplexity scores.
To minimize spam, one approach is to identify generic sentences or invitations to share articles on social networks commonly found at the end of documents. We create a list of frequently used words associated with these paragraphs and then filter out paragraphs that contain an excessive proportion of words from this list.
To augment our ability to identify non-human-generated content, we consider a subset of 10 million documents from OSCAR (Ortiz Suárez et al., 2020), a web-crawled corpus. We extract the words from these documents, removed punctuations, converted them to lowercase, and retain only the words occurring at least twice, which we refer to as common words. We filter out paragraphs with a too low common word ratio.
The detail of the cutoff values for all text filters at the paragraph level is present in Table 3.

By applying these node-level and paragraph-level filters, we ensure that only high-quality and relevant images and paragraphs are retained for further processing and analysis.

**Document-level filtering**   For document-level filtering, we start by removing all documents with no images or with more than 30 images. We have found that when there are too many images in a document, they are often not related to each other, and are more likely to be considered as spam.
For text filters, we use the same filters as for filtering at paragraph level. Since we are at the document level, the filter metrics are more precise, and we can typically set stricter cutoff values while limiting the number of false positives. The cutoff values used are also present in Table 3.

After these filtering steps, we obtained 365 million web documents and 1.4 billion images (potentially duplicated in different documents at this stage).

### A.1.5   Additional Filtering and Deduplication Steps

**Exclusion of opted-out images**   To respect the preferences of content creators, we remove all images for which creators explicitly opted out of AI model training. We used the Spawning API[9] to verify that the images in the dataset respect the original copyright owners' choices. This step had a small impact on the overall dataset, by removing only 0.047% of the images.

---

[9] https://api.spawning.ai/spawning-api

| Metric | Cutoff type | Cutoff value (paragraph-level) | Cutoff value (document-level) |
|---|---|---|---|
| Number of words | min | 4 | 10 |
| Number of words | max | 1,000 | 2,000 |
| Character repetition ratio | max | 0.1 | 0.1 |
| Word repetition ratio | max | 0.1 | 0.2 |
| Special character ratio | max | 0.3 | 0.275 |
| Stop word ratio | min | 0.3 | 0.35 |
| Flagged word ratio | max | 0.01 | 0.01 |
| Punctuation ratio | min | 0.001 | 0.03 |
| Spam word ratio | max | 0.12 | 0.12 |
| Common word ratio | min | 0.8 | 0.9 |
| Language identification prediction score | min | 0.8 | 0.8 |
| Perplexity score | max | 1500 | 1500 |

Table 3: Cutoff values for text filters at paragraph and document levels. A 'min' (or 'max') cutoff indicates that any paragraph or document, depending on the level, with a value for the considered metric strictly below (or above) the cutoff value is removed.

**Image deduplication based on URL**    Prior to this step, it is possible for the same image to be present in multiple documents under the same URL. However, we observe that the distribution of image occurrences was highly skewed, with the majority of images appearing only once, while a small subset of images appeared hundreds of thousands of times. Upon closer examination, we notice that these frequently occurring images are predominantly comprised of common advertisements encountered during the crawling process, browser-specific icons, and similar elements. To address this issue, we remove all images that appear more than 10 times across the entire dataset. This approach significantly reduces the presence of unwanted images. We intentionally do not perform strict deduplication, as we observe that when an image is duplicated only a few times across different documents, the surrounding text and contextual information tend to vary. These diverse contexts associated with the duplicated image could be beneficial for the training of a model. We also deduplicate images within the same document.

**NSFW image removal**    We use an open-source NSFW classifier[10] to reduce the proportion of explicit adult content within our dataset. We carefully choose a cutoff that reduces as much as possible the proportion of false positives. Indeed, if favoring precision to recall may seem to be a good idea to remove as much undesirable content as possible, it hurts diversity. An analysis of false positives shows that in many cases, simple portrait photos of women are classified as pornographic, which is not the case for men. People of color are also more often misclassified. We remove the entire document when a pornographically classified image is found in the document. In addition, we also remove all images whose URLs contain the sub-strings *porn*, *sex* or *xxx*. We remove approximately 1% of the documents with this filter. Note that many pornographic documents have been previously removed by the filter on flagged words.

**Document deduplication based on URL**    Since we consider many Common Crawl dumps, it is possible that several documents may be associated with the same URL, despite the initial deduplication efforts. Recognizing the inherent similarity among these documents, we opt to retain only the most recent document for each common URL.

**Document deduplication based on set of images**    It is possible that documents with different URLs and domain names are very similar and have not been removed by the first

---

[10] https://github.com/GantMan/nsfw_model

deduplication, for instance, news articles copied and pasted multiple times across various sources. To mitigate this, we form groups of documents with an identical set of images, and we keep only the most recent document for each group.

**Paragraph deduplication across documents of the same domain names**   To eliminate generic spam phrases commonly found at the end of documents, such as "Share on Facebook," "Post a comment," or "Accept the cookies," we implement a paragraph-level deduplication process within documents sharing the same domain name. This approach aims to enhance the quality of the text by removing redundant and repetitive content. For each domain name, we identify paragraphs that appear at least three times in an identical manner across associated documents. These repetitive paragraphs are subsequently removed from the documents, resulting in the elimination of approximately 15% of the text present in the web documents.

After all these steps, the final dataset contains 141 million documents and 353 million images, of which 298 million are unique.

We observe that using stricter values for the filtering steps yields fewer multimodal documents, although not of higher quality. As such, we invite users who are interested in manipulating a smaller subset of OBELICS to start with a random subset.

## A.2 Analysis of `OBELICS`

## A.2.1 Examples of Multimodal Web Documents

**Document**

Right now, in Costa Rica, the classic dry season has been evasive. As the sky clouds over just as it did during June, and the rains begin to fall, it almost feels like the whole usual dry season thing has been waived. Cold fronts continue to arrive and subsequently douse the country with Atlantic showers while a "Nina" effect over in the Pacific has only added to the wet situation. Despite the umbrella test, there are good things associated with this. High biodiversity is correlated with high rainfall and that makes for more birds. It's one of the main reasons why so many species occur in Costa Rica.

It can be a challenge to find them under varying degrees of precipitation but what's a birder gonna do? It's part of the local birding scene and when the clouds take a lunch break, the birds suddenly come out to play. Get enough of those breaks and you can get into some stellar birding, especially when high rainfall earlier in the year encouraged the trees and bushes to grow lots of bird friendly fruit. Seriously, it's a smorgasbord out there right now, the tanagers, manakins, thrushes, trogons, and toucans are going to feed whether it rains or not.

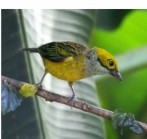

When the sun eventually does come out, there seem to be certain birds that take advantage of the sudden bloom of warmth and UV rays. Yesterday morning at El Tapir, a client and myself bore witness to what can happen when the rain finally comes to a stop and the sun, unhindered by clouds, punctuates the sky. At first, there was little activity, as if the birds were still numbed by the constant falling of water, still in denial that the rain had stopped. A few wrens and some other birds vocalized, a pair of Mealy Parrots fluttered overhead but pretty quiet otherwise. However, while the birds of the forest slowly came back to life, the Rufous-tailed Hummingbirds were racing around the garden. Judging by their frantic behavior (even for hummingbirds), it seemed like they hadn't eaten quite enough in days. Or maybe they just didn't get their fill of nectar? Whatever the case, they were drinking from the Verbena flowers as if they were participants in some avian Bacchus festivities. Unfortunately, they didn't invite any other hummingbirds to the party and took great efforts to bounce any potentially crashing woodnymph, Snowcap, or Violet-headed.

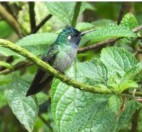

Dressed for the party, still denied entrance. Name's not down, not coming in.

It took a while but the Rufous-taileds seemed to eventually get their fill (or became too inebriated) and as the sun took over the garden space, a couple other hummingbird species braved the post party scene. One of the most cooperative was a male Black-crested Coquette.

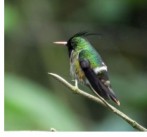

As is typical with coquettes, the male chose to perch on a bare twig for extended periods of time before carefully flying down to drink from the Verbena. Much to our satisfaction, this particular exquisite beauty preferred to feed on a bush right in front of us.

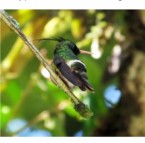

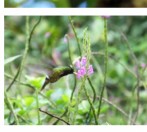

It was interesting to note that as the coquette fed, the Rufous-taileds seemed to be more concerned with chasing a female woodnymph and a Violet-headed Hummingbird. It was as if they didn't notice the coquette as the smaller hummingbird slowly moved in and out of the flowering bushes, pumping its tail up and down the entire time.

As we enjoyed the coquette show, a few raptors eventually took advantage of thermals created by the sun to fly high over the garden.

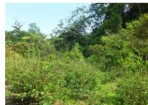

As it turned out, the Black-crested Coquette was just the headliner for the main act.

The first on stage was an adult Ornate Hawk-Eagle. It called so loudly, I expected to see it floating just over the canopy but no, it was already high above the forest, fooling the eyes into thinking they were seeing something as small as an Accipiter or a dainty kite. The eagle called over and over, it was as if it couldn't help itself, singing because it could finally soar up and reach those heights again after a repressive bout of cool weather and constant rain. Alive again! Like there was nothing else in its world, it yelled into the skies above the forest, fluttered its wings and made shallow dives, displaying over a busy road for all who felt like peering into the high blue sky. Once, I swear it did a barrel roll, vocalizing the entire time.

As the eagle continued with its expression of exuberant defiance, next on the list were a pair of Barred Hawks. These broad-winged, short-tailed raptors gave their gull-like vocalizations as they soared into view. They continued to make circles up above the forest until they reached a point where they also began to display by soaring in tandem, calling the entire time.

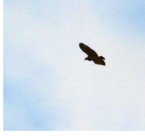

One of the Barred Hawks, looks like it found some food that morning.

While this raptor fest was going on, a pair of King Vultures also soared into view, not as close as the hawks but still within eyeshot to appreciate their bold, black and white pattern. They seemed to be displaying as well, one bird almost flying into the other one and then close tandem flight, like the other raptors, taking advantage of a beautiful, new day.

It might rain a lot but it eventually stops. When it does, the sun's coming out something good is going to happen, the time comes for action. Whether you be a Spizaetus or a birder, be ready to make your move and catch the lightbridge found in that window of respite.

Figure 7: Example of a document in `OBELICS`.
From http://birdingcraft.com/wordpress/2018/01/23/what-happens-with-birding-in-costa-rica-when-the-rain-stops/

**Document**

Can I Expect Compensation For My Injuries?

The word "compensation" can be a touchy issue when discussing personal injuries and settlement. Even when it is the sole objective of a lawsuit or some other legal proceeding, mentioning compensation for my injuries can create false expectations in someone's mind if not addressed in the proper context. A San Diego lawyer who practices personal injury law, for example, says that it is crucial to ensure that a person seeking compensation has the right mindset and expectations whenever such cases are discussed. If mishandled, it can lead to anger and resentment on their part.

After suffering injuries in an accident, whether at the workplace or through some other negligent action, seeking damages is understandably a logical thing to do. Such legal action may entail going to court and making your case known to the judge. If there's a large sum of money involved, one should always prepare for a protracted legal battle.

The truth is that both a trial and an outright settlement can have very different variables and outcomes. Choosing to go to trial might seem like a good option. After all, many culpable parties are usually in a more agreeable frame of mind once the threat of a court case looms, making them more likely to offer a settlement.

Such parties usually settle a case out of self-interest. The strain and financial cost of sustaining an effective legal defense can be ruinous. In many cases, though, insurance companies step in to offer compensation. After all, many employers and other parties like vehicle drivers tend to have insurance coverage for exactly those sorts of situations. After sustaining injuries, an amount of money is offered to the victim to help them with medical bills and any other expenses they may have incurred due to injuries sustained. Many liable parties and insurance companies usually prefer a quick out-of-court settlement because court cases can become an expensive affair.

As a victim, it is always prudent to remember that a court case could be decided against you, thereby leaving you with no compensation at all. While some cases usually result in higher dollar amounts being doled out as a settlement because of successful litigation, many victims do not want to take the risk. Such victims are already drowning in medical bills by the time they think of seeking compensation for their injuries. That's why most prefer a swift settlement if given the option.

How An Insurance Provider Chooses To Settle A Claim

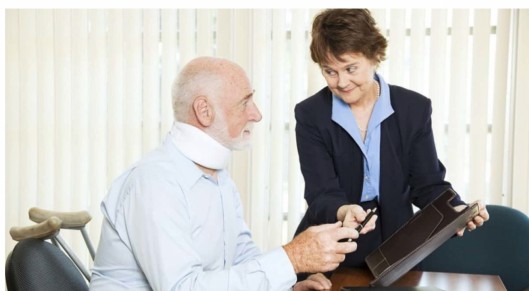

As mentioned, an insurance provider involved in such cases would rather settle a personal injury case out of court. A jury trial is risky for both the personal injury victim and the insurance provider. The unpredictability of many such cases means that an insurance carrier could find themselves having to fork out significantly higher amounts of money in compensation than if they had chosen a quick, out-of-court settlement.

An insurance provider is always looking to minimize its costs while ensuring less risk. As such, they may opt to compensate a personal injury victim while simultaneously seeking reimbursement from the third party that is responsible for your injuries, usually from such a third party's insurance carrier.

It's crucial to remember that, in some jurisdictions, an insurance provider is entitled to a percentage of your compensation if they already settled your medical bills prior to you receiving the settlement. This amount is commensurate with all your medical expenses.

There now exist online settlement calculators that purport to provide a rough estimate of the compensation a personal injury victim can expect. You put in the various numerical values and factors related to your case, and the site will give you a general idea of what to expect in monetary terms. However, sometimes this information can be misleading and hence you should never rely on it. Even with the best personal injury lawyers handling your case, it is difficult if not impossible to account for all of the numerous variables. Even in cases with admitted liability of a third party, getting a sense of a definitive dollar amount for compensation is still difficult. The extent of the injury suffered, emotional distress and pain, and loss of potential future earnings are things that can prove very tricky to quantify. As such, it is inadvisable to rely on online settlement calculators for such estimates.

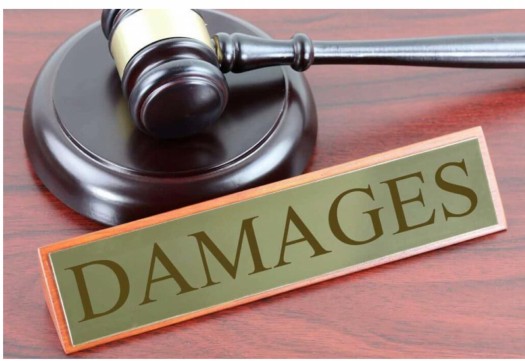

Medical costs and other expenses related to economic losses due to the injury are factored into calculating the damages awarded to a personal injury victim. Loss of companionship, deprived enjoyment of life, and emotional distress are some of the issues that determine compensation but may be hard to nail down.

While seemingly straightforward, any compensation awarded to a victim only happens after consideration of all relevant factors. Sometimes, the victim of personal injury is to blame, whether partly or in full. This has the potential to negate any compensation or at least diminish it. An experienced personal injury attorney can help such victims to fully understand all the different scenarios involved in such cases.

Can A Victim Reject A Settlement Offer?

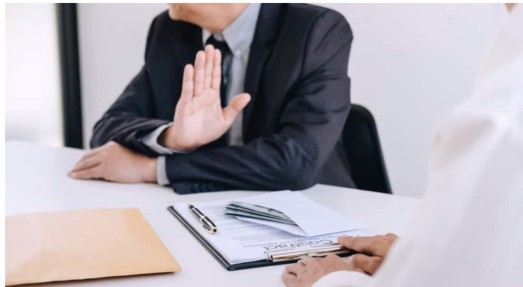

A personal injury victim is well within his rights to reject compensation. This could arise when the victim feels that the alleged guilty party has not put forward a dollar amount that is representative of the extent of injury and loss incurred. As a victim, you can sit down with your personal injury attorney to get a sense of how such scenarios generally play out. The accused party may be doing this intentionally, hoping that the victim accepts this offer without much consideration. You can express dissatisfaction with such an offer through a personal injury demand letter, outlining your grievances and why you believe you are entitled to more.

In a nutshell, a victim is entitled to compensation when the accused party is found to be responsible for the accident that caused injury to the victim. With many variables in such cases, there is no minimum amount of money set as the standard for compensation. Each case is examined on the merits of its unique factors, ensuring an equitable settlement for all parties.

Figure 8: Example of a document in `OBELICS`.
From 

## Document

The Marvel Cinematic Universe has created some magnificent things over the last decade and a half. This cinematic universe has brought them back from the cusp of bankruptcy and into times of abundance once again. The success of the MCU has now allowed Marvel Studios to bring out the obscure characters from comic pages onto the silver screen. Who would have thought that Kit Harrington would be playing Dane Whitman in the MCU? It is relevant because Dane Whitman will become Black Knight, the greatest swordsman on the planet who fights alongside Avengers.

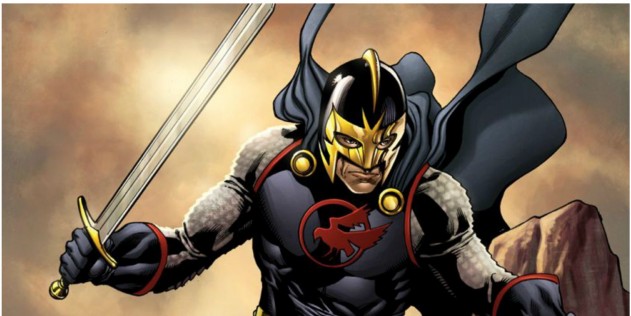

Who is this Black Knight? Why do we care? And why are we talking about this after a movie about cosmic beings like the Eternals and the Celestials? Does a sword not seem moot in front of infinite cosmic energy? Not when it is this sword. You see, in the after-credits scene of Eternals, Dane Whitman aka the love interest of Sersi unveils a sword. This sword seems to whisper to him and looks like the cursed Ebony Blade from the comics. Dane Whitman in the comics wields this blade and calls himself the Black knight, a superhero who assists the Avengers in various battles.

But there is a catch. The Ebony Blade was supposed to be welded by the pure of heart as explained by Merlin who created the sword. But the secret of the sword is that it can only be wielded by those who are impure of heart. The blade was actually designed by Merline for Sir Percy ( ancestor of Dane Whitman) to make him the greatest swordsman at the time. But the catch is that the blade seeks out evil inside you and amplifies it until there is nothing but a berserker left.

This seems to be true in the MCU too. The Ebony Blade blesses its user with incredible power, but it also comes at an incredible cost. This sword also prolongs its user's life as much as it can. The last Black Knight before Dane Whitman was Nathan Garrett, his uncle who is mentioned in the movie several times. This Black Knight was a villain who was defeated by the Avengers in the comics. But here, he is nowhere to be seen. There is a reason for this and the reason is most likely that Nathan Garrett will work better as a villain against Dane Whitman than the Avengers of the MCU.

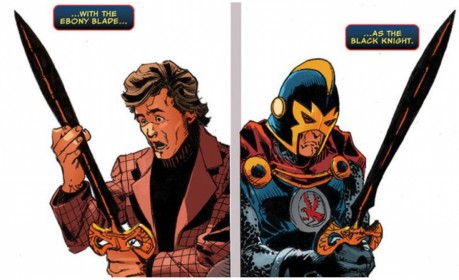

This Ebony Blade is a malicious piece of weaponry. It was created by Merline so that Sir Percy may sully his honor in battle but it also gave him immense power in the series. There is a possibility that we will see a similar story play out with Kit Harrington's character in the MCU. Moreover, there is another question that we must address. Who does the voice at the end of the second after-credits scene belong to? It has been confirmed by Chloe Zhao that it is Mahershala Ali's Blade who has come to recruit Dane.

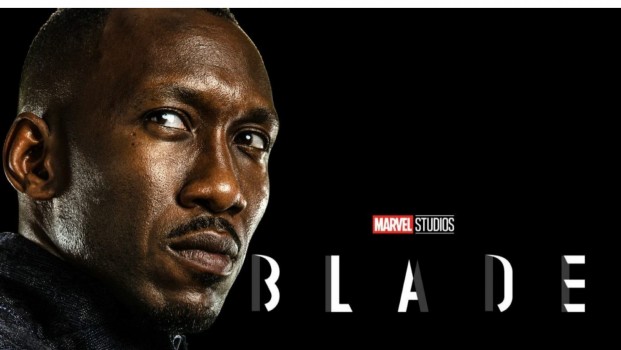

Blade was the iconic movie that popularised superhero vampire hunters but there is another element to this hero that connects to the Black Knight. The Excaliburs was a team that got together to fight against supernatural foes. One of these foes was Dracula himself who was the one who created a replica of the Ebony Blade. In the comics, it was revealed that the Ebony Blade wielded by Dane was actually the replica created by Dracula.

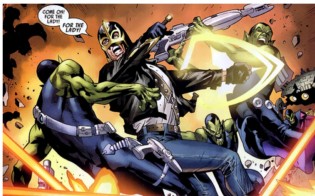

This made the Blade itself vampiric in some sense and if this storyline is kept intact in the MCU then it won't be surprising to see Dane in Blade. It seems obvious at this point that the Ebony Blade will soon be replaced with Excalibur in the movies. Thena plays with the original King Arthur sword in the Domo in Eternals. This is confirmed by sprite. We think that Dane will try to use the Ebony Blade to try to rescue Sersi from Arishem but would be asked by Blade to help him. This would start the Excalibur team-up and lead to the events of Blade where they hunt down Dracula.

After this, Dane might be consumed by the evil within the Ebony Blade and would discard it. To make sure that he can continue to be the hero he needs to be he will be given the Excalibur from The Domo and he will become the true leader of this new team. We think this will be the logical progression of events, taking a note from the current lineup of MCU movies, unless more are announced. Let us know what you think about this in the comments below and keep watching this space for everything Marvel, DC, and Hollywood. Excelsior!!!

Figure 9: Example of a document in OBELICS. From https://www.quirkybyte.com/blog/2021/11/how-dane-whitman-will-become-black-knight-kit-harringtons-character-explained/

## A.2.2 Unwanted Document Containing Many Images

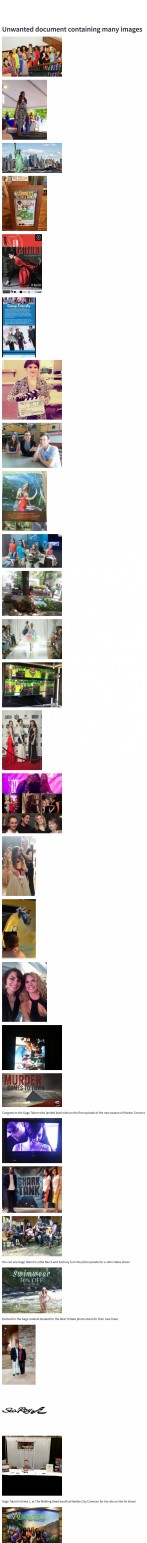

Figure 10: Undesirable document containing many images. Text is only present in small proportions, and the relation between the images is not always clear.

### A.2.3 Top 100 Domains

| Rank | Domain name | Number of documents |
|------|-------------|---------------------|
| 1 | www.dailymail.co.uk | 434,498 |
| 2 | en.wikipedia.org | 155,258 |
| 3 | nypost.com | 141,494 |
| 4 | www.thestar.com | 138,224 |
| 5 | sputniknews.com | 133,695 |
| 6 | www.rediff.com | 133,233 |
| 7 | www.theepochtimes.com | 132,539 |
| 8 | www.fool.com | 125,220 |
| 9 | www.businessinsider.com.au | 123,841 |
| 10 | www.bustle.com | 122,581 |
| 11 | www.dailysabah.com | 120,029 |
| 12 | www.firstpost.com | 119,642 |
| 13 | www.irishtimes.com | 118,329 |
| 14 | theathletic.com | 101,982 |
| 15 | www.news.com.au | 98,339 |
| 16 | www.indiatimes.com | 98,197 |
| 17 | www.theglobeandmail.com | 92,805 |
| 18 | tvtropes.org | 92,104 |
| 19 | www.dailydot.com | 91,034 |
| 20 | mashable.com | 88,310 |
| 21 | observer.com | 87,336 |
| 22 | www.cbsnews.com | 86,759 |
| 23 | www.rappler.com | 86,554 |
| 24 | www.tmz.com | 84,472 |
| 25 | www.salon.com | 84,420 |
| 26 | www.modernghana.com | 83,918 |
| 27 | www.foxnews.com | 83,002 |
| 28 | www.huffpost.com | 81,701 |
| 29 | www.ndtv.com | 81,549 |
| 30 | www.thisismoney.co.uk | 80,930 |
| 31 | www.famousbirthdays.com | 78,931 |
| 32 | www.engadget.com | 76,817 |
| 33 | www.rnz.co.nz | 76,327 |
| 34 | www.metro.us | 75,627 |
| 35 | www.patheos.com | 75,003 |
| 36 | www.news24.com | 73,883 |
| 37 | www.thestar.com.my | 73,265 |
| 38 | www.dw.com | 72,774 |
| 39 | www.npr.org | 71,939 |
| 40 | koreajoongangdaily.joins.com | 71,091 |
| 41 | peoplesdaily.pdnews.cn | 71,048 |
| 42 | pagesix.com | 70,602 |
| 43 | www.thenigerianvoice.com | 70,470 |
| 44 | wikimili.com | 69,928 |
| 45 | www.indiebound.org | 67,986 |
| 46 | www.cricketcountry.com | 66,605 |
| 47 | expressdigest.com | 64,250 |
| 48 | www.capitalfm.co.ke | 64,163 |
| 49 | www.bizpacreview.com | 64,157 |
| 50 | www.wionews.com | 63,797 |
| 51 | profootballtalk.nbcsports.com | 63,532 |
| 52 | jamaica-gleaner.com | 63,137 |
| 53 | www.rte.ie | 63,074 |

| | | |
|---|---|---|
| 54 | www.aspentimes.com | 62,552 |
| 55 | kids.kiddle.co | 62,419 |
| 56 | english.alarabiya.net | 60,368 |
| 57 | www.jellypages.com | 59,381 |
| 58 | people.com | 59,293 |
| 59 | muse.jhu.edu | 59,061 |
| 60 | www.geeky-gadgets.com | 58,975 |
| 61 | www.khaleejtimes.com | 58,851 |
| 62 | www.nbcsports.com | 57,922 |
| 63 | en.topwar.ru | 56,723 |
| 64 | www.thewrap.com | 56,146 |
| 65 | www.outlookindia.com | 55,752 |
| 66 | www.celebdirtylaundry.com | 55,618 |
| 67 | time.com | 55,527 |
| 68 | www.dailystar.co.uk | 55,503 |
| 69 | www.legit.ng | 55,395 |
| 70 | www.thehansindia.com | 55,109 |
| 71 | www.bbc.co.uk | 55,015 |
| 72 | newsinfo.inquirer.net | 54,927 |
| 73 | nesn.com | 54,756 |
| 74 | www.tellerreport.com | 53,939 |
| 75 | www.rawstory.com | 53,676 |
| 76 | www.thestatesman.com | 53,286 |
| 77 | wccftech.com | 52,510 |
| 78 | forward.com | 51,969 |
| 79 | nationalinterest.org | 51,851 |
| 80 | www.pearltrees.com | 50,933 |
| 81 | www.contactmusic.com | 50,284 |
| 82 | www.tweaktown.com | 50,138 |
| 83 | www.destructoid.com | 50,081 |
| 84 | www.publishersweekly.com | 49,735 |
| 85 | www.cbs58.com | 49,680 |
| 86 | www.markedbyteachers.com | 48,994 |
| 87 | www.caughtoffside.com | 48,857 |
| 88 | www.islamicinvitationturkey.com | 48,721 |
| 89 | dailyhive.com | 48,447 |
| 90 | www.aljazeera.com | 47,393 |
| 91 | www.bbc.com | 47,349 |
| 92 | worldbulletin.dunyabulteni.net | 47,300 |
| 93 | www.romper.com | 47,115 |
| 94 | www.catchnews.com | 47,025 |
| 95 | www.odt.co.nz | 46,712 |
| 96 | www.jewishpress.com | 46,688 |
| 97 | www.irishcentral.com | 46,629 |
| 98 | techcrunch.com | 46,539 |
| 99 | www.nhl.com | 46,247 |
| 100 | www.tuko.co.ke | 46,106 |

Table 4: Ranking of the 100 domains with the highest number of associated documents in OBELICS.

### A.2.4 Topic Modeling with 20 Topics

| Concept | Ratio | Related words |
|---|---|---|

| Justice | 5.16% | said, police, people, year, according, court, case, told, news, man, two, death, also, one, old, investigation, found, fire, officers |
|---|---|---|
| Politics | 6.35% | said, state, government, would, president, trump, law, court, party, public, new, election, states, political, federal, house, people, also, bill |
| Family | 5.24% | family, one, day, back, life, time, home, would, old, said, years, like, two, love, mother, children, first, man, went |
| Music | 5.23% | music, album, band, song, new, songs, show, also, first, sound, rock, one, musical, year, released, live, festival, record, track |
| Climate | 3.46% | water, energy, climate, species, also, earth, space, one, used, gas, use, solar, natural, power, carbon, years, change, system, may |
| Business | 7.12% | year, company, million, market, said, new, business, companies, per, also, billion, percent, price, financial, money, industry, years, growth, according |
| Sports | 3.75% | game, season, team, first, year, two, said, three, play, last, games, one, win, second, points, coach, back, players, four |
| Sports (2nd) | 5.67% | team, first, year, season, league, last, two, club, world, race, one, game, win, time, back, players, match, second, final |
| Automotive | 4.18% | new, car, also, design, one, power, cars, two, model, use, used, system, camera, first, speed, engine, high, vehicle, battery |
| Cinema | 7.36% | film, story, series, movie, book, new, show, one, also, characters, character, first, world, star, films, love, best, life, man |
| War | 4.26% | war, country, said, military, countries, russia, world, russian, government, united, international, people, states, president, also, security, israel, army, forces |
| Gaming | 5.77% | game, use, also, new, games, data, one, users, app, online, using, video, google, players, play, time, used, information, content |
| Health | 3.0% | health, also, may, medical, patients, disease, study, people, treatment, cancer, body, use, drug, research, risk, brain, care, virus, cases |
| Food | 2.08% | food, also, one, beer, like, eat, made, wine, restaurant, make, coffee, meat, well, used, tea, sugar, use, water, taste |
| Urban | 4.62% | city, area, new, park, one, building, town, road, also, north, day, around, river, island, south, place, along, local, two |
| Existence | 5.23% | one, people, god, life, world, women, many, even, human, may, like, way, men, often, would, man, also, social, power, must |
| Asia | 1.61% | india, indian, also, china, said, chinese, government, minister, pakistan, country, delhi, kong, hong, people, singh, two, khan, sri, asia |
| History | 4.24% | book, art, first, history, years, new, century, work, one, books, also, church, american, world, time, museum, english, known |
| Education | 5.11% | school, said, students, work, university, new, community, also, people, years, year, education, program, women, working, support, college, children, project |
| Other | 10.56% | like, one, get, would, time, people, really, know, even, think, much, good, going, way, see, could, make, want, things, something |

Table 5: LDA with 20 topics, trained on 100,000 random web documents. A concept for each topic is derived from the related words.

### A.2.5  Topic Modeling with 200 Topics

| Concept | Ratio | Related words |
|---|---|---|
| Celebrity Relationships | 0.52% | star, fans, show, love, instagram, couple, together, shared, relationship, revealed, year, kim, charlie, told, actress, pete, new, former, old, lisa |
| Music Industry | 1.47% | band, music, song, album, songs, rock, tour, live, singer, show, record, country, bands, released, stage, one, love, played, pop |
| Racial Diversity | 0.26% | black, white, people, race, african, american, racial, community, racism, gay, racist, americans, diversity, lgbtq, justice, color, lgbt, gender, discrimination, queer |
| Language Usage | 0.17% | language, english, word, words, name, languages, use, used, text, names, letter, letters, meaning, translation, writing, spoken, speech, speaking, speak, term |
| Team Spirit | 0.38% | said, get, team, good, really, going, lot, year, think, got, great, like, last, back, well, play, time, guys, big, hard |
| News Media | 0.28% | news, media, radio, fox, press, magazine, journalists, television, journalism, story, newspaper, editor, journalist, coverage, times, broadcast, interview, daily, podcast, show |
| European Culture | 0.04% | van, dutch, netherlands, tattoo, amsterdam, belgium, portugal, belgian, der, tattoos, portuguese, bulgaria, sofia, holland, bulgarian, lisbon, santos, europe, tulip, brussels |
| European Nations | 0.19% | european, germany, german, europe, berlin, sweden, poland, greece, also, countries, swedish, polish, czech, denmark, norway, austria, greek, hungary, finland |
| Film Industry | 1.29% | film, movie, films, director, movies, best, actor, hollywood, documentary, cinema, role, screen, story, directed, production, actors, also, oscar, award |
| Australian Achievements | 0.12% | australia, australian, new, zealand, sydney, award, melbourne, awards, year, victoria, queensland, south, nsw, brisbane, australians, best, won, auckland, prize |
| Culinary Delights | 0.88% | cream, recipe, cheese, make, chocolate, made, bread, add, taste, ice, butter, sauce, cake, sugar, cook, food, salt, milk, sweet |
| Life and Death | 0.4% | death, one, people, life, world, dead, even, lives, many, die, died, lost, killed, still, never, man, end, left, day, hope |
| Spiritual Philosophy | 0.2% | philosophy, spiritual, buddhist, religion, religious, yoga, buddha, meditation, buddhism, tibetan, guru, book, practice, knowledge, thought, mind, life, modern, texts, tradition |
| Cultural Histories | 0.13% | jewish, jews, indigenous, native, holocaust, rabbi, tribe, people, indian, community, peoples, tribal, israel, tribes, anti, culture, land, camp, history, torah |
| Personal Development | 0.07% | says, people, explains, like, new, adds, get, work, want, also, tells, lot, say, year, years, really, working, part, wants, help |

| | | |
|---|---|---|
| Royal Families | 0.23% | king, prince, royal, queen, princess, charles, henry, elizabeth, duke, harry, palace, meghan, family, william, anne, castle, kate, lady, diana, edward |
| Daily News | 0.19% | said, week, friday, monday, wednesday, according, tuesday, thursday, news, last, day, told, sunday, saturday, reported, statement, days, morning, hours |
| Creative Projects | 0.19% | project, design, work, working, projects, creative, create, idea, team, process, also, ideas, new, make, designer, created, started, concept, worked, wanted |
| Legal Investigations | 0.6% | investigation, information, former, report, fbi, department, office, according, documents, evidence, public, intelligence, government, claims, allegations, corruption, fraud, alleged, officials, federal |
| Medical Procedures | 0.19% | surgery, skin, pain, treatment, cancer, procedure, patients, teeth, bone, patient, surgical, injury, eye, hair, tissue, surgeon, tooth, breast, honey, medical |
| Athletic Competitions | 0.46% | olympic, sports, world, athletes, games, sport, olympics, gold, team, medal, NUMm, event, won, year, championships, competition, athlete, time, first |
| Historical Artifacts | 0.62% | ancient, century, NUMth, history, temple, stone, roman, years, one, city, also, greek, found, known, built, old, site, time, today |
| Literary Works | 0.87% | book, books, read, story, author, novel, writing, reading, series, stories, first, written, fiction, published, readers, characters, world, one, write, new |
| Time Progression | 0.73% | one, year, years, last, still, could, even, time, big, new, two, much, like, back, next, would, since, another, well, already |
| Everyday Life | 0.2% | day, time, sleep, night, home, hours, room, water, house, bed, days, morning, work, get, every, food, hour, two, camp, minutes |
| Colorful Nature | 0.16% | color, tea, dark, white, green, flowers, skin, like, black, flower, colors, blue, rose, leaves, light, pink, also, red, used, golden |
| Automotive Industry | 1.21% | car, cars, engine, vehicle, new, vehicles, model, electric, ford, drive, also, wheel, rear, speed, driving, toyota, motor, front, power |
| American Cities | 0.11% | new, york, california, city, san, los, angeles, francisco, chicago, jersey, state, times, diego, brooklyn, center, santa, bay, seattle, county |
| Political Movements | 0.57% | political, people, power, party, government, right, america, politics, anti, war, state, world, left, free, nation, democracy, american, country, media, system |
| Mythical Creatures | 0.12% | bear, wolf, dragon, snake, bears, lion, like, tiger, monster, wild, human, wolves, animals, snakes, cave, creatures, giant, humans, hunter, dragons |
| Asian Cultures | 0.09% | north, korea, harry, kim, korean, potter, south, jon, thrones, jong, pyongyang, stewart, nuclear, ron, warner, hogwarts, house, game, colbert, peninsula |
| Data Modeling | 0.31% | data, model, number, value, using, numbers, function, used, models, values, two, example, method, figure, one, set, problem, object, line |
| Romantic Stories | 1.34% | story, love, life, girl, one, new, woman, find, young, man, finds, characters, father, friend, two, character, family, romance, secret, series |

| | | |
|---|---|---|
| Medical Research | 0.41% | cancer, cells, cell, dna, disease, gene, human, patients, genetic, immune, protein, treatment, genes, bacteria, researchers, diseases, research, proteins, study, clinical |
| Fitness and Training | 0.21% | running, race, run, training, marathon, fitness, miles, exercise, bike, mile, runners, NUMk, course, gym, finish, cycling, yoga, half, runner |
| Personal Perspectives | 1.43% | like, people, think, really, would, know, going, get, see, one, lot, things, something, time, want, way, much, thing, say, could |
| Gastronomy Scene | 0.44% | food, restaurant, coffee, bar, restaurants, menu, chef, chicken, pizza, meal, kitchen, dishes, dinner, eat, dining, burger, table, meals, served, like |
| Labor Rights | 0.29% | workers, work, employees, job, jobs, union, pay, labor, working, employment, insurance, employers, wage, employee, company, paid, worker, labour, staff, business |
| Competitive Sports | 0.75% | game, second, goal, first, ball, half, back, minutes, win, lead, two, points, score, minute, final, match, side, three, time |
| Public Events | 0.71% | year, event, festival, christmas, day, events, NUMth, show, night, tickets, special, holiday, party, live, celebrate, held, also, place, saturday |
| Digital Marketing | 0.37% | digital, content, marketing, media, brand, advertising, platform, online, campaign, ads, business, industry, social, new, users, platforms, brands, companies, internet, consumers |
| Public Safety | 0.24% | safety, report, action, letter, statement, said, incident, ban, made, public, actions, claims, reported, according, response, taken, complaints, following, take, serious |
| French Heritage | 0.1% | french, france, paris, jean, saint, les, des, pierre, dame, marie, europe, macron, notre, louis, european, michel, jamaica, jacques, emmanuel |
| Eastern European Politics | 0.38% | russian, russia, ukraine, ukrainian, moscow, putin, soviet, state, vladimir, war, azerbaijan, country, armenian, armenia, president, russians, union, sanctions, region |
| Horror Entertainment | 0.58% | movie, story, horror, characters, character, film, action, one, plot, ghost, scene, evil, movies, like, series, original, genre, dark, scenes, first |
| Political Campaigns | 1.25% | trump, president, election, vote, campaign, obama, party, biden, house, donald, political, republican, presidential, voters, democratic, democrats, candidate, clinton, candidates, white |
| Indian Cinema | 0.64% | film, khan, actor, also, movie, bollywood, films, kapoor, indian, actress, seen, role, singh, india, release, hindi, kumar, directed, hai, salman |
| Corporate Leadership | 0.82% | years, board, director, president, team, business, leadership, work, executive, also, chief, role, member, management, service, experience, served, staff, working |
| Law Enforcement | 1.94% | police, said, officers, man, officer, arrested, year, old, incident, two, found, according, investigation, killed, department, shot, scene, vehicle, suspect |
| Football Clubs | 1.26% | club, league, season, united, premier, players, city, football, chelsea, team, arsenal, player, manchester, liverpool, game, side, back, last, games |

| | | |
|---|---|---|
| Essential Skills | 0.84% | get, make, need, one, also, time, best, want, many, use, may, take, find, like, even, help, way, good, people, much |
| Artistic Expression | 0.75% | art, museum, artist, work, artists, exhibition, painting, works, gallery, arts, paintings, collection, artistic, drawing, new, show, contemporary, painted, artwork |
| American Regions | 0.22% | state, county, texas, florida, north, south, michigan, ohio, carolina, states, virginia, west, georgia, center, university, washington, colorado, iowa, arizona |
| Industrial Production | 0.28% | production, company, industry, mining, manufacturing, gold, mine, port, supply, project, companies, factory, industrial, plant, steel, products, equipment, coal, goods |
| Global Affairs | 0.36% | world, countries, international, united, trade, china, states, global, country, foreign, europe, region, asia, economic, european, nations, south, india, east |
| Government Affairs | 1.26% | minister, government, said, meeting, party, president, prime, would, members, committee, council, parliament, also, general, decision, agreement, political, secretary, national, commission |
| Software Development | 0.67% | code, use, file, using, software, version, files, windows, run, server, application, web, source, open, user, system, new, linux, install |
| UK Happenings | 0.22% | london, british, england, britain, centre, brexit, bbc, wales, labour, west, manchester, johnson, north, programme, south, across, may, year, east |
| Real Estate Market | 0.16% | property, housing, estate, home, real, homes, house, rent, properties, market, land, mortgage, rental, sale, houses, price, owner, buyers, sales, units |
| Fashion Trends | 0.43% | fashion, hair, wearing, dress, wear, look, style, clothing, clothes, black, wore, designer, beauty, shirt, women, also, made, show, costume, new |
| Gaming Culture | 0.38% | game, cards, card, games, play, players, poker, player, casino, online, gambling, win, deck, playing, betting, lottery, bet, slot, chess, played |
| Famous Personalities | 0.04% | bond, kelly, martin, daniel, peter, doctor, tony, johnny, parker, sean, evans, frank, andy, ian, lucas, dave, reynolds, spy, emily, amber |
| Wildlife Conservation | 0.61% | species, birds, bird, animals, fish, found, animal, also, wild, wildlife, eggs, habitat, large, food, like, small, humans, insects, many, endangered |
| Pandemic Responses | 0.94% | covid, pandemic, health, people, virus, coronavirus, vaccine, cases, said, spread, outbreak, public, lockdown, vaccines, government, new, disease, vaccination, deaths |
| Popular Names | 0.11% | john, michael, david, paul, jones, james, johnson, mike, jim, steve, robert, two, bob, davis, moore, allen, brian, mark, one |
| Christian Theology | 0.45% | god, jesus, christ, bible, christian, church, faith, lord, people, gospel, paul, christians, john, prayer, word, biblical, kingdom, pastor, moses |
| Sports | 0.77% | season, team, game, nba, games, basketball, players, player, play, coach, league, hockey, points, teams, nhl, played, first, star, year |
| Cybersecurity | 0.63% | data, security, network, internet, cloud, information, access, technology, services, service, NUMg, software, computer, systems, networks, cyber, devices, users, attacks, use |

| | | |
|---|---|---|
| Business/Finance | 0.78% | company, business, companies, market, industry, investment, investors, capital, tech, firm, ceo, based, technology, billion, businesses, group, million, financial, growth |
| Professional Wrestling | 0.18% | wwe, ring, wrestling, match, rick, randy, champion, title, wrestler, vince, show, fans, wrestlers, owens, tag, baker, triple, shane, raw, cody |
| Japanese Culture/Tech | 0.15% | anime, musk, japanese, tesla, manga, series, elon, japan, ninja, episode, samurai, kai, characters, demon, karate, character, also, dragon, arc, tokyo |
| Scottish Personalities | 0.03% | brown, scotland, scottish, gordon, glasgow, celtic, perry, walker, murray, graham, letter, edinburgh, cover, campbell, watson, thomas, also, well, neil, henderson |
| Streaming Media | 0.12% | video, youtube, videos, live, watch, channel, streaming, audio, content, stream, channels, footage, shows, online, also, NUMk, recording, watching, clip, one |
| Christianity | 0.36% | church, catholic, pope, religious, christian, churches, bishop, francis, faith, holy, priest, saint, mass, vatican, religion, pastor, christ, parish, christians |
| Smartphone Technology | 0.83% | phone, apple, samsung, iphone, pro, smartphone, device, galaxy, camera, also, display, battery, new, sNUM, screen, NUMgb, phones, NUMg, android |
| Urban Development | 0.78% | city, project, area, council, residents, community, park, town, street, public, local, cities, new, development, mayor, urban, construction, district, building |
| Sociocultural Issues | 0.39% | social, culture, society, cultural, people, political, different, moral, identity, important, values, issues, often, public, role, many, way, community, understanding, view |
| Common Male Names | 0.03% | smith, jack, tom, ben, adam, alex, kevin, richard, simon, holmes, billy, bell, oliver, harvey, jake, collins, burke, baldwin, joel, aaron |
| Combat Sports | 0.49% | fight, title, tennis, champion, ufc, round, world, boxing, fighter, one, win, open, martial, first, match, mma, fighters, fighting, career |
| Indian Politics | 0.64% | india, indian, state, delhi, government, also, minister, bjp, said, modi, singh, chief, congress, crore, pradesh, mumbai, gandhi, lakh, hindu |
| Military History | 0.25% | war, world, battle, empire, british, army, history, german, peace, great, military, wars, end, conflict, power, two, land, forces, soldiers, fight |
| Internet Cartography | 0.04% | www, map, sri, http, https, maps, lanka, com, atlas, derby, tamil, lankan, html, maria, angelo, tara, colombo, org, mapping, easter |
| European Football | 0.46% | league, champions, team, goals, world, season, football, club, cup, madrid, barcelona, player, real, players, match, messi, ronaldo, liverpool, final |
| Mobile Applications | 0.73% | app, google, apple, android, users, mobile, apps, phone, new, devices, device, ios, iphone, microsoft, use, also, features, user, screen, windows |
| Korean Entertainment | 0.11% | lee, korean, korea, kim, south, park, seoul, drama, group, bts, jin, jung, first, also, members, won, woo, hyun, young, min |
| Economics | 1.01% | market, price, prices, markets, growth, inflation, economy, stock, economic, rate, rates, investors, higher, year, demand, stocks, trading, dollar, gold |

| | | |
|---|---|---|
| Video Games | 0.49% | games, game, xbox, gaming, nintendo, video, play, console, playstation, mario, psNUM, one, sony, players, steam, gamers, switch, playing, titles |
| Time Indicators | 0.3% | first, years, since, time, two, NUMth, three, total, day, year, may, second, september, june, january, november, four, NUM/NUM, april |
| Science Fiction/Fantasy | 0.14% | star, wars, trek, lego, luke, figures, force, series, jedi, kirk, toy, universe, figure, new, ship, galaxy, crew, fans, space, disney |
| Music Production | 1.09% | album, sound, music, band, track, song, guitar, metal, sounds, tracks, songs, record, bass, vocals, new, release, rock, like, released, drums |
| Transportation | 0.42% | document, token, road, end, replaced, bike, traffic, driving, drivers, bus, train, driver, bridge, car, station, ride, roads, route, transport, rail |
| Personal Life | 1.14% | life, people, love, world, many, time, one, always, years, great, every, like, way, friends, never, day, work, first, hope, best |
| American History | 0.6% | american, history, NUMs, new, first, years, century, america, early, states, united, NUMth, became, world, many, one, today, time, war |
| Global Policy | 0.96% | change, climate, development, economic, government, global, policy, need, sector, world, public, new, support, economy, national, social, future, health, impact, crisis |
| South Asian Affairs | 0.2% | pakistan, afghanistan, taliban, kashmir, bangladesh, khan, india, pakistani, afghan, also, nepal, country, indian, kabul, jammu, singh, islamabad, ali, lahore, karachi |
| Sports Scores | 0.83% | game, points, first, season, two, three, win, second, four, team, lead, run, third, one, five, scored, home, games, point |
| Travel/Daily Life | 1.03% | day, time, back, get, last, one, got, good, night, next, morning, went, first, trip, week, see, around, way, little |
| Announcements | 0.83% | new, year, first, last, time, next, NUMth, month, also, release, announced, two, months, march, since, october, september, week, may |
| Online Dating | 0.13% | dating, gay, online, sites, date, site, tinder, free, men, best, matchmaking, meet, guy, hookup, guys, app, apps, relationship, singles, dates |
| Superhero Comics | 0.42% | comic, marvel, comics, man, batman, spider, superhero, character, avengers, superman, universe, hero, captain, new, heroes, fans, issue, super, characters, also |
| Space Exploration | 0.31% | space, nasa, mission, mars, drone, launch, rocket, satellite, robot, earth, robots, drones, moon, first, station, orbit, satellites, spacecraft, technology |
| Musical Performance | 0.57% | music, jazz, musical, concert, piano, orchestra, composer, musicians, classical, symphony, played, performance, playing, performed, piece, work, instruments, also, festival, instrument |
| Personal Finance | 0.17% | money, pay, card, credit, bank, cash, vegas, payment, paid, account, las, payments, fees, cost, cards, amount, buy, service, fee |
| Television Shows | 0.74% | show, series, season, episode, netflix, shows, episodes, television, comedy, watch, cast, fans, also, new, seasons, character, drama, viewers, first |

| | | |
|---|---|---|
| Celebrity Culture | 0.11% | taylor, jackson, justin, swift, star, jennifer, singer, jay, tyler, cohen, nicole, spencer, also, eddie, cole, carrie, amy, lopez, bieber, casey |
| Environmental Conservation | 0.32% | water, river, land, environmental, forest, wildlife, conservation, area, natural, lake, areas, project, environment, rivers, dam, resources, forests, national, management |
| Physical/Quantum Sciences | 0.35% | water, air, chemical, used, process, material, surface, materials, quantum, temperature, high, oxygen, carbon, radiation, particles, liquid, salt, energy, pollution, chemicals |
| Astronomy | 0.37% | earth, sun, moon, planet, sky, stars, solar, star, space, light, universe, planets, telescope, years, scientists, system, galaxy, eclipse, dark |
| Islamic/Middle Eastern Culture | 0.19% | muslim, saudi, muslims, islam, islamic, arabia, egypt, arab, dubai, allah, uae, ali, middle, abu, prophet, religious, muhammad, mosque, iran, egyptian |
| Gender Issues | 0.14% | women, men, woman, female, girls, gender, male, abortion, sexual, girl, young, sex, life, equality, feminist, man, violence, ladies, rights, boys |
| Fantasy/Mythology | 0.03% | sam, lewis, max, rings, twin, troy, monkey, toy, stephen, palmer, doll, hobbit, tolkien, zeus, lord, monkeys, seth, horse, toys, witch |
| Video Game Mechanics | 0.36% | attack, damage, enemy, pokemon, use, weapon, enemies, level, also, fight, battle, attacks, players, power, weapons, ability, magic, hero, character, armor |
| MMORPG Gaming | 1.16% | game, games, players, play, new, player, world, playing, characters, gameplay, mode, character, also, story, battle, fun, experience, free, fantasy |
| Energy and Environment | 0.65% | energy, oil, gas, power, carbon, solar, fuel, emissions, electricity, climate, wind, renewable, coal, natural, green, production, industry, fossil, environmental |
| Financial Regulations | 0.57% | tax, financial, bank, government, debt, income, banks, money, taxes, budget, economy, finance, loan, pay, billion, loans, credit, economic, fund |
| US Legislation | 0.75% | state, bill, would, federal, house, senate, congress, law, legislation, act, states, governor, government, passed, public, committee, lawmakers, plan, funding |
| Subjective Experience | 0.91% | like, good, really, one, well, much, great, bit, even, little, quite, also, though, still, pretty, lot, see, get, better, would |
| Parenthood | 0.16% | children, child, kids, parents, baby, age, young, birth, parent, pregnancy, pregnant, family, families, babies, adults, mother, old, early, mothers |
| Personal Experiences | 1.93% | like, get, one, know, got, really, good, little, even, think, guy, thing, going, love, pretty, right, let, much, never, back |
| Education | 0.55% | school, students, education, schools, college, student, high, university, class, teachers, year, teacher, campus, program, learning, teaching, classes, children, grade, parents |
| Latin American Cultures | 0.17% | mexico, spanish, italian, spain, italy, san, mexican, latin, puerto, del, cuba, rico, colombia, costa, america, cuban, venezuela, juan, country |

| | | |
|---|---|---|
| Technological Systems | 0.68% | system, new, technology, systems, development, also, use, time, process, high, based, performance, work, used, well, using, provide, quality, level, developed |
| Social Movements | 0.6% | rights, people, government, human, violence, protest, freedom, police, country, protests, law, civil, political, protesters, movement, state, justice, activists, right, groups |
| Surfing/Beach Culture | 0.02% | scott, ryan, wilson, joe, anderson, wave, josh, sarah, phil, surf, jackie, waves, robinson, logan, beach, ken, surfing, phoenix, duncan, gibson |
| Brazilian Culture | 0.03% | brazil, brazilian, miller, rio, phillips, paulo, portuguese, peterson, grande, são, janeiro, ivy, bolsonaro, herman, silva, state, amazon, sao, spike, hernandez |
| Literature/Poetry | 0.32% | poetry, writing, essay, writer, poem, poems, literary, literature, work, poet, book, published, writers, wrote, write, english, works, collection, written, life |
| Family Life | 0.58% | family, years, wife, home, mary, born, school, life, funeral, friends, died, church, death, service, many, member, may, mrs, passed |
| Cricket | 0.47% | cricket, india, test, match, runs, team, england, series, first, wickets, ipl, overs, game, tNUM, played, indian, ball, innings, captain |
| Canadian/Irish Affairs | 0.09% | canada, canadian, ireland, irish, toronto, ontario, vancouver, dublin, province, alberta, northern, canadians, ottawa, montreal, provincial, centre, quebec, north, trudeau |
| Music Industry | 1.01% | music, album, song, artists, artist, hip, single, hop, released, new, songs, rapper, track, video, rap, pop, release, hit, singer |
| Criminal Justice | 0.6% | prison, crime, criminal, court, charges, sexual, trial, case, jail, years, crimes, guilty, victims, murder, abuse, accused, sentence, justice, convicted |
| Academic Research | 0.66% | university, research, science, professor, institute, studies, college, scientific, school, work, study, engineering, national, international, department, students, degree, academic, center |
| Names and Dates | 0.02% | williams, hill, ross, carter, kennedy, clark, jan, nelson, jordan, stanley, rated, murphy, arthur, marshall, hudson, feb, nov, oct, mar |
| Weather Conditions | 0.49% | weather, ice, snow, mountain, winter, north, temperatures, cold, climate, south, high, lake, rain, temperature, east, west, summer, conditions, ski |
| Health and Medicine | 0.54% | blood, brain, disease, symptoms, may, heart, patients, body, treatment, also, cause, risk, pain, condition, effects, common, severe, doctor, pressure |
| Cryptocurrency | 0.47% | bitcoin, blockchain, crypto, cryptocurrency, digital, mining, ethereum, cryptocurrencies, currency, exchange, btc, market, network, tokens, users, price, nft, trading, transactions, token |
| Diet and Nutrition | 0.38% | food, diet, weight, health, body, fat, eating, foods, eat, sugar, healthy, also, high, diabetes, people, meat, protein, obesity, levels |
| Actions and Movements | 0.12% | back, get, time, take, right, move, way, next, see, start, around, keep, make, end, away, going, one, left, another, turn |

| | | |
|---|---|---|
| Historic Landmarks | 0.36% | NUMth, town, village, name, william, george, century, hall, john, family, built, castle, early, house, mill, street, history, became, morris |
| Electronic Devices | 0.41% | power, light, battery, use, control, device, used, system, led, also, using, devices, high, signal, air, electrical, switch, low, sensor |
| Performing Arts | 0.43% | theatre, show, dance, stage, play, theater, performance, production, audience, musical, opera, arts, broadway, dancing, cast, performances, performing, company, ballet, shakespeare |
| Mental Health | 0.26% | mental, people, health, disorder, depression, help, self, anxiety, stress, emotional, person, life, physical, may, often, brain, also, social, autism, feel |
| Online Interaction | 0.35% | post, blog, read, comments, posted, like, would, one, see, com, please, know, article, share, site, email, comment, posts, link, page |
| Substance Usage | 0.27% | drug, drugs, cannabis, marijuana, use, cbd, medical, effects, addiction, fda, used, alcohol, cocaine, substance, prescription, heroin, treatment, products, thc, also |
| Outdoor Landscapes | 0.46% | tree, trees, trail, water, road, river, along, forest, area, around, small, park, one, near, old, wood, way, hill, across, ground |
| Colors | 0.06% | red, blue, white, green, black, yellow, color, light, flag, orange, grey, colors, gray, logo, one, pearl, hat, look, colour, two |
| Israel and Fishing | 0.19% | israel, israeli, fish, palestinian, jerusalem, fishing, gaza, palestinians, netanyahu, hamas, jewish, bank, west, palestine, state, arab, israelis, trout, salmon |
| Air Travel | 0.4% | airport, flight, aircraft, air, airlines, plane, flights, travel, airline, passengers, aviation, flying, fly, international, airports, pilot, passenger, boeing, service |
| Waste and Recycling | 0.16% | plastic, waste, made, used, use, bags, make, bag, paper, items, nike, fabric, shoes, cola, using, coca, trash, recycling, also, shoe |
| Philosophical Discourse | 0.34% | would, even, one, could, however, much, fact, yet, rather, far, though, many, well, might, perhaps, less, long, despite, may, time |
| Problems and Issues | 0.16% | could, problem, many, may, problems, due, however, issues, issue, would, even, also, cause, result, still, time, situation, damage, impact, without |
| Firearms and Malaysia | 0.17% | gun, shooting, guns, malaysia, hunting, rifle, firearms, shot, deer, weapons, shoot, weapon, malaysian, pistol, firearm, ammunition, rmNUM, hunt, buck |
| Disney and Animation | 0.12% | disney, magic, world, ray, animation, alice, walt, park, animated, fairy, ride, parks, disneyland, theme, magical, pixar, jungle, studios, orlando, characters |
| Middle Eastern Conflict | 0.81% | syria, turkey, forces, iraq, military, security, attacks, attack, killed, syrian, terrorist, turkish, war, people, state, group, isis, terrorism, terrorists, government |
| Physical Descriptions | 0.48% | eyes, like, face, could, head, hand, back, little, looked, hands, said, around, look, body, would, voice, see, away, hair, felt |
| Architecture | 0.62% | building, house, room, space, built, floor, construction, wall, buildings, new, home, design, tower, two, walls, architecture, roof, rooms, designed |

| | | |
|---|---|---|
| Travel Destinations | 0.94% | city, hotel, park, one, visit, tour, world, town, place, travel, area, many, also, trip, beautiful, places, visitors, located, island |
| Computer Hardware | 0.41% | intel, performance, computer, memory, amd, core, graphics, usb, windows, laptop, drive, cpu, card, power, nvidia, hardware, gpu, processor, gaming |
| African Nations | 0.17% | africa, south, african, kenya, country, cape, uganda, rNUM, zimbabwe, continent, national, congo, africans, west, tanzania, president, town, johannesburg, rwanda, nairobi |
| Military Operations | 0.37% | military, army, war, soldiers, forces, troops, general, service, battle, soldier, commander, men, armed, corps, force, command, training, unit, guard, combat |
| Tobacco and Cookies | 0.15% | cookies, website, smoking, use, tobacco, cigarettes, buy, smoke, experience, cigar, cookie, necessary, used, ivermectin, cigarette, consent, online, may, vaping, also |
| Nigerian Politics | 0.67% | state, nigeria, said, government, nigerian, governor, president, ghana, lagos, buhari, also, nNUM, nigerians, country, national, federal, people, apc, security, abuja |
| Family Dynamics | 0.54% | family, father, mother, son, old, daughter, home, children, years, year, parents, wife, young, brother, life, dad, two, house, sister |
| Farming and Agriculture | 0.4% | plant, farmers, farm, food, plants, agriculture, garden, soil, agricultural, seeds, grow, growing, seed, crop, crops, production, farming, farms, fruit, harvest |
| Retail Industry | 0.27% | store, market, products, sales, amazon, stores, customers, price, company, business, retail, product, buy, shop, online, consumers, brand, shopping, sell, selling |
| Online Resources | 0.32% | download, information, free, page, available, online, book, edition, website, pdf, article, site, published, library, content, please, text, may, read |
| Personal Experiences | 2.07% | would, time, could, one, didn, first, back, got, went, years, came, wanted, made, started, took, never, day, wasn, thought, even |
| Theology and Morality | 0.45% | god, man, one, lord, world, life, earth, upon, power, may, spirit, human, evil, love, heaven, gods, soul, must, every, shall |
| Sports and Games | 1.29% | season, game, team, football, nfl, yards, baseball, games, players, league, coach, field, play, year, player, bowl, quarterback, teams, first |
| Asia and Pacific | 0.07% | japan, japanese, tokyo, vietnam, indonesia, pacific, hawaii, island, vietnamese, indonesian, islands, asian, also, asia, west, rice, jakarta, abe, hawaiian |
| Healthcare | 0.27% | health, care, medical, hospital, patients, doctors, healthcare, patient, treatment, services, medicine, doctor, hospitals, hiv, nursing, nurses, emergency, insurance, nurse, staff |
| Commemorations | 0.21% | day, memorial, anniversary, national, NUMth, ceremony, veterans, flag, honor, statue, cemetery, people, nation, war, country, president, service, years, monument |
| Collectibles and Auctions | 0.32% | gold, collection, silver, watch, auction, box, original, sold, coin, coins, one, made, sale, watches, design, set, edition, also, rare |

| | | |
|---|---|---|
| East Asia | 0.18% | china, chinese, kong, hong, singapore, philippines, beijing, taiwan, thailand, shanghai, asia, also, thai, province, asian, country, philippine, city, manila |
| Maritime Exploration | 0.4% | sea, island, ship, boat, ocean, water, coast, beach, bay, ships, marine, islands, boats, cruise, port, waters, crew, fishing, sailing |
| Natural Disasters | 0.39% | fire, people, storm, hurricane, disaster, emergency, fires, damage, flood, earthquake, rescue, smoke, flooding, firefighters, homes, residents, burning, hit, area |
| Legal Matters | 0.69% | court, law, case, judge, legal, supreme, justice, decision, attorney, filed, trial, cases, courts, lawyer, lawyers, lawsuit, appeal, ruling, judges |
| Dimensions and Positioning | 0.47% | two, side, one, top, right, back, cut, line, use, small, used, hand, like, left, body, front, size, using, around |
| Relationships and Marriage | 0.18% | marriage, sex, relationship, married, wedding, love, couple, sexual, divorce, man, husband, wife, couples, together, woman, partner, men, one, relationships, bride |
| Community Projects | 0.84% | community, support, group, people, members, program, help, local, foundation, event, also, work, organization, part, project, together, youth, young, year |
| Photography | 0.26% | image, camera, images, photo, photos, NUMd, photography, pictures, cameras, picture, light, lens, photographer, capture, photographs, taken, shot, look, using, shoot |
| Competitive Sports | 0.88% | team, players, teams, cup, tournament, world, football, competition, final, round, golf, play, club, match, first, won, league, win, sports |
| Innovation and Science | 0.57% | world, human, new, reality, create, like, time, life, future, nature, work, experience, way, process, space, ideas, different, form, idea, science |
| Personal Opinions | 1.87% | people, know, like, think, say, even, want, make, one, something, things, someone, way, doesn, would, good, need, person, feel, never |
| Statistics | 0.99% | percent, per, year, number, according, cent, average, report, increase, years, rate, million, data, population, last, people, increased, growth, higher |
| Personal Communication | 0.15% | said, would, told, people, added, could, asked, also, going, think, want, year, last, say, saying, one, interview, make, come, according |
| Animal Companions | 0.3% | dog, dogs, cat, animals, animal, cats, horse, pet, breed, horses, pets, also, owner, bull, owners, pig, rescue, puppy, pigs, humans |
| Scientific Research | 0.41% | study, research, data, researchers, found, results, studies, risk, analysis, evidence, group, published, test, findings, based, university, likely, may, could |
| Mystery and Adventure | 0.43% | man, back, one, left, door, street, front, around, away, saw, car, went, two, night, told, heard, took, later, behind, another |
| Motor Racing | 0.85% | race, racing, team, season, track, car, races, second, first, win, championship, lap, two, driver, top, series, year, drivers, fNUM |
| International Politics | 0.56% | united, states, iran, border, trump, nuclear, president, immigration, security, country, administration, foreign, american, countries, migrants, policy, refugees, immigrants, government, washington |

| | | |
|---|---|---|
| Air Defense | 0.34% | air, aircraft, force, military, navy, defense, defence, wing, fighter, missile, flying, base, naval, command, pilot, pilots, flight, forces, jet |
| Additional Information | 0.62% | within, however, additionally, stated, mentioned, one, extra, password, might, individuals, simply, time, present, actually, get, place, may, together, different |
| Financial Performance | 0.62% | million, year, billion, company, quarter, sales, revenue, per, said, share, total, according, last, first, NUMm, percent, expected, growth, reported |
| Alcohol and Beverages | 0.38% | beer, wine, drink, alcohol, brewery, drinking, wines, bottle, brewing, beers, craft, taste, brew, drinks, whisky, ale, tasting, bar, whiskey, bottles |
| Celebrity Profiles | 0.66% | also, career, born, known, years, worth, age, net, life, famous, american, became, name, first, million, started, year, appeared, actress |
| Storytelling and Narratives | 1.26% | like, life, story, world, one, time, sense, way, yet, much, work, makes, narrative, every, often, takes, moments, something, stories, piece |
| Legislation | 0.78% | law, act, rules, may, legal, laws, government, public, must, state, regulations, would, information, rule, commission, states, required, order, authority |
| Social Media | 0.45% | twitter, facebook, social, media, instagram, post, people, account, also, pic, tweet, share, news, online, posted, video, users, page, wrote, shared |
| Comparative Analysis | 0.42% | one, also, however, two, may, different, many, used, example, well, often, first, part, although, another, time, known, fact, various, number |

Table 6: LDA with 200 topics, trained on 100,000 random web documents. A concept for each topic is derived from the related words.

## A.3  Ethical discussion

At the beginning of the project, we reflected on ethical principles[11] guiding the project, including the creation of the dataset, in order to incorporate ethical values we agreed on. These values motivated the careful crafting of the content filters. For instance, we used the Spawning API to respect as much as possible the consent decisions of content creators or iterated significantly on filters around pornographic content.

Exploring large-scale corpora is often a tedious process which contributes to the lack of transparency and lack of documentation around these artifacts. With that in mind, we built an interactive visualization[12] of OBELICS which allows browsing through a subset (11M documents) of the dataset and navigate the different topics covered. Yet, we note that despite our efforts, OBELICS contains a small proportion of documents that are not suitable for all audiences. For instance, one might find the cluster named "Sex" which predominantly contains descriptions of pornographic movies along with pornographic images. Other clusters would contain advertising for sex workers, or reports of violent shootings. In our experience, these documents represent a small proportion of all the documents.

Due to the nature of our dataset (multimodal documents extracted from the web), OBELICS inherits the same ethical concerns of unlabeled text corpora crawled from the web: difficulty to document/inspect, presence of unintended biases, under-representation of certain demographics, etc. These concerns have been well documented for text corpora (Biderman and Scheirer, 2020; Bender et al., 2021). Data audits have shed light on the some limitations and unintended biases contained in these text corpora (Caswell et al., 2020; Dodge et al., 2021). The augmentation of text corpora with interleaved images is a recent development of multimodal machine learning. We hope that our dataset along with exploration tools will serve as a solid ground for endeavors such as data audits. Existing works auditing large-scale multimodal datasets have focused on image-text pairs datasets (Birhane et al., 2021) and highlight how curation and filtering decisions lead to biases (including racism and misogyny) in the resulting pairs. We believe that interleaved image-text datasets will play a significant role in the development of increasingly more capable multimodal models, and having large-scale versions of these datasets that are transparent, maintained and in open-access is critical.

We also have evaluated the trained models as part of a red-teaming effort and a systematic evaluation of the generations produced by the model compared across the axis of gender and race. More specifically, the model was separately prompted to write a resume, a dating profile, and a headline about a person's recent arrest based on their appearance. We studied the generations and analyzed the trends for each protected characteristic using FairFace (Kärkkäinen and Joo, 2021) and StableBias (Luccioni et al., 2023). The details of these evaluations and insights are made public as part of the model release. As an example, the model trained on OBELICS associates men more frequently than women with terms like "financial", "development", "product", and "software".

## A.4  Building the Model

### A.4.1  Architecture Details

We closely follow the Flamingo architecture introduced in Alayrac et al. (2022). To form the model, we combine a pre-trained image encoder, a pre-trained language model, and add newly initialized parameters of the form of Perceiver blocks (Jaegle et al., 2021) and Transformer-based cross-attentions blocks inserted within the language model every 4 layers.

The pre-trained backbones are frozen during the training, and only the new parameters are updated along with the embeddings of additional tokens.

Following Dehghani et al. (2023), we apply a layer normalization on the projected queries and keys of both the Perceiver and cross-attention blocks, which improved training stability

---

[11] https://huggingface.co/blog/ethical-charter-multimodal
[12] https://atlas.nomic.ai/map/f2fba2aa-3647-4f49-a0f3-9347daeee499/ee4a84bd-f125-4bcc-a683-1b4e231cb10f

in our early experiments. We use the RMSNorm implementation (Zhang and Sennrich, 2019) for the layer normalization.

| Total | Trainable | Language Model | Vision Model | Perceiver | Cross-Attentions |
|-------|-----------|----------------|--------------|-----------|------------------|
| 9B | 1.5B | 7B | 630M | 126M | 1.4B |
| 80B | 14B | 65B | 630M | 126M | 13.9B |

Table 7: Breakdown of model parameters. We use LLaMA (Touvron et al., 2023) for the language backbone and OpenCLIP (`https://laion.ai/blog/large-openclip/`) for the vision backbone.

### A.4.2 Training Details

We roughly use the same set hyper-parameters for all the runs presented in Figure 6 and Table 2, as detailed in Table 8. The training of `IDEFICS` uses a larger batch size and examples of longer sequence length. In all experimental runs, we employ the AdamW optimizer (Loshchilov and Hutter, 2017) and incorporate an auxiliary loss, denoted as $z\_loss = 10^{-3} \times log^2(Z)$, to encourage the softmax normalizer $log(Z)$ to get closer to 0 (Chowdhery et al., 2022). We use gradient clipping of 1.0.

During the training, two models – `IDEFICS` and the 9B-parameter model trained on `LAION + OBELICS` – encountered unrecoverable loss spikes. As a remedial measure, we restarted the training from a checkpoint before the spike, shuffled the data and optionally reduced the learning rate. Both models underwent exactly three restarts within the training duration.

The four runs conducted have distinct data mixtures as detailed in Table 10, and Tabel 9 gives the number of tokens and images in the different datasets. Each run involves training on a mixture of web documents and image-text pairs. A sampling probability $p$ determines the mixture of these two data sources, which influences the frequency of batches originating from web documents versus those from image-text pairs.

For `IDEFICS` and `IDEFICS-9B`, the web-document dataset includes both `OBELICS` and Wikipedia, and the image-text pair dataset included `LAION` and Public Multimodal Dataset (PMD) (Singh et al., 2022). Given Wikipedia and PMD's higher quality but lower number of examples, we repeat PMD three times and Wikipedia three times.

We used a deduplicated version of `LAION` (Webster et al., 2023) for all the runs where this dataset was used.

### A.4.3 Compute Details

We train the 9B-parameter models on `OBELICS`-only and `LAION`-only on 32 80GB A100 GPUs, and on `OBELICS + LAION` on 64 80GB A100s, for approximately 6 days. These 3 trainings have the same effective batch size. We train `IDEFICS` on 512 80GB A100 GPUs and `IDEFICS-9B` on 128 80GB A100 GPUs for about 14 days each. The compute infrastructure is hosted on an AWS cluster located in Oregon.

### A.4.4 Evaluation

To ensure fair comparisons against Flamingo (Alayrac et al., 2022), we make sure that we are using the same evaluation splits for each benchmark. We evaluate the models using an in-context learning approach (Brown et al., 2020), with random in-context examples. For the 0-shot evaluations, as in Alayrac et al. (2022), we use 2 random priming in-context examples but without passing the associated images. We systematically use different data splits to select the best-performing prompt (which involves creating validation sets from the training sets, following the methodology proposed by Alayrac et al. (2022)). Table 11 lists the prompts used for each model and task.

For the classification tasks (HatefulMeme (Kiela et al., 2020), IIIT-5k (Mishra et al., 2012)), we use rank classification, i.e. we compute the log probability of the prompt followed by

| | Parameters | IDEFICS-80B | IDEFICS-9B |
|---|---|---|---|
| **Perceiver Resampler** | *Number of Layers* | 6 | 6 |
| | *Number of Latents* | 64 | 64 |
| | *Number of Heads* | 16 | 16 |
| | *Resampler Head Dimension* | 96 | 96 |
| **Model** | *Language Model Backbone* | Llama-65b | Llama-7b |
| | *Vision Model Backbone* | `laion/CLIP-ViT -H-14-laion2B -s32B-b79K` | `laion/CLIP-ViT -H-14-laion2B -s32B-b79K` |
| | *Cross-Layer Interval* | 4 | 4 |
| **Training** | *Sequence Length* | 1024 | 1024 |
| | *Effective Batch Size (# of tokens)* | 3.67M | 1.31M |
| | *Max Training Steps* | 200K | 200K |
| | *Weight Decay* | 0.1 | 0.1 |
| | *Optimizer* | Adam(0.9, 0.999) | Adam(0.9, 0.999) |
| | *Gradient Clipping* | 1.0 | 1.0 |
| | *Z-loss weight* | 1e-3 | 1e-3 |
| **Learning Rate** | *Initial Max* | 5e-5 | 1e-5 |
| | *Initial Final* | 3e-5 | 6e-6 |
| | *Decay Schedule* | Linear | Linear |
| | *Linear warmup Steps* | 2K | 2K |
| **Large-scale Optim.** | *Gradient Checkpointing* | True | True |
| | *Precision* | Mixed-pres bf16 | Mixed-pres bf16 |
| | *ZeRO Optimization* | Stage 3 | Stage 3 |

Table 8: Training Hyper-Parameters

| Data Source | Data Type | # Tokens in Source | # Images in Source | Epochs |
|---|---|---|---|---|
| OBELICS | Unstructured Multimodal Web Documents | 114.9B | 353M | 1 |
| Wikipedia | Unstructured Multimodal Web Documents | 3.192B | 39M | 3 |
| LAION | Image-Text Pairs | 29.9B | 1.120B | 1 |
| PMD | Image-Text Pairs | 1.6B | 70M | 3 |

Table 9: Number of tokens and images in the different datasets used for the training of IDEFICS.

each of the labels individually, and select as the predicted label the one with the highest probability.

| Model | OBELICS | Wikipedia | LAION | PMD |
|---|---|---|---|---|
| 9B-parameter model, OBELICS + LAION | 50% | 0% | 50% | 0% |
| 9B-parameter model, OBELICS only | 100% | 0% | 0% | 0% |
| 9B-parameter model, LAION only | 0% | 0% | 100% | 0% |
| IDEFICS-9B | 73.85% | 6.15% | 17.18% | 2.82% |
| IDEFICS | 73.85% | 6.15% | 17.18% | 2.82% |

Table 10: Breakdown of the dataset mixtures used. Percentages correspond to the effective number of tokens seen from each dataset.

For the image captioning (COCO (Lin et al., 2014), Flickr30k (Young et al., 2014)) and visual question answering tasks (VQAv2 (Antol et al., 2015), OKVQA (Marino et al., 2019), TextVQA (Singh et al., 2019), VizWiz (Gurari et al., 2018)), we report evaluation in the open-ended setup. We use the greedy decoding as we found that it increased the performance. However, we observe that the models tend to generate long answers. To truncate the generated caption or answer, unless specified otherwise, we use a list of manually selected stop words. For VisDial, since the evaluation metric is NDCG, we instead rank the possible candidates for each question.

The VQA tasks comporting a high proportion of questions with a single-word answer, it was beneficial for the 9B-parameter model trained on LAION only to keep the first word of the generated answer as the prediction to boost its performance.

| Task | Model | Prefix prompt | Example prompt | Stop words |
|---|---|---|---|---|
| VQAv2 OKVQA TextVQA | IDEFICS IDEFICS-9B 9B LAION only 9B OBELICS only 9B LAION + OBELICS | {bos_token}Instruction: provide an answer to the question. Use the image to answer.\n | Image:{token_around_image}{image_token}{token_around_image}Question: {question} Answer: {answer}\n | "Question", "User", "Image", "task", "What", "Who", "When", "Where", "Why", "How" |
| COCO Flickr30k | IDEFICS IDEFICS-9B 9B OBELICS only 9B LAION + OBELICS | {bos_token} | Image:{token_around_image}{image_token}{token_around_image}Caption: {caption}\n | "Caption", "Description", "User", "Image", "task" |
| COCO Flickr30k | 9B LAION only | {bos_token}Instruction: provide a short caption of the input image.\n | Image:{token_around_image}{image_token}{token_around_image}Image description: {caption}\n | "Caption", "Description", "User", "Image", "task" |
| Hateful-Memes | IDEFICS IDEFICS-9B 9B LAION only 9B OBELICS only 9B LAION + OBELICS | It's a conversation between a human, the user, and an intelligent visual AI, Bot. The user sends memes with text written on them, and Bot has to say whether the meme is hateful or not. | {token_around_image}{image_token}{token_around_image}is an image with written "{context}" on it. Is it hateful? Answer: {class_name} | ✗ |
| IIIT5k | 9B LAION only 9B OBELICS only 9B LAION + OBELICS | ✗ | {token_around_image}{image_token}{token_around_image}"{class_name}" is written on the picture. | ✗ |
| VizWiz | IDEFICS IDEFICS-9B | {bos_token}Task: Answer the questions based on the image when possible, otherwise say unanswerable.\n | Image:{token_around_image}{image_token}{token_around_image}Question: {question} Answer: {answer}\n | "Question", "User", "Image", "task", "What", "Who", "When", "Where", "Why", "How" |
| VisDial | IDEFICS IDEFICS-9B | ✗ | {token_around_image}{image_token}{token_around_image}{caption}. {context}{class_name}. | ✗ |

Table 11: We select the prompts from a pool of candidates by evaluating 5 intermediate checkpoints on the query and support validation task sets. To form the prompt with $N$ priming examples, we concatenate the prefix prompt, followed by $N$ example prompts filled with data from the priming examples, and finally the example prompt filled with data from the example to be evaluated. The data to be replaced is between curly brackets.

### A.4.5 Additional Experimental Results

In Figure 11, we plot the performance per benchmark for the 9B-parameter models trained on `LAION` only, `OBELICS` only, and a mixture of `OBELICS` and `LAION`. We notice that, even if the training on `LAION` only is smooth and the loss keeps decreasing (there are no spikes nor instabilities), performance starts to decrease after a certain point on visual question answering benchmarks. We hypothesize that training on image-text pairs can allow a fast association of concepts between images and texts, but fails to teach the model more complex reasoning skills required to solve visual question answering. We tried many different prompt candidates in order to boost the performance of the model trained on `LAION` only for the VQA tasks, without much success.

On the other hand, we note that training on image-text pairs yield stronger performance on image captioning tasks than on multimodal documents only. This is expected since training and evaluation correspond to the exact same task.

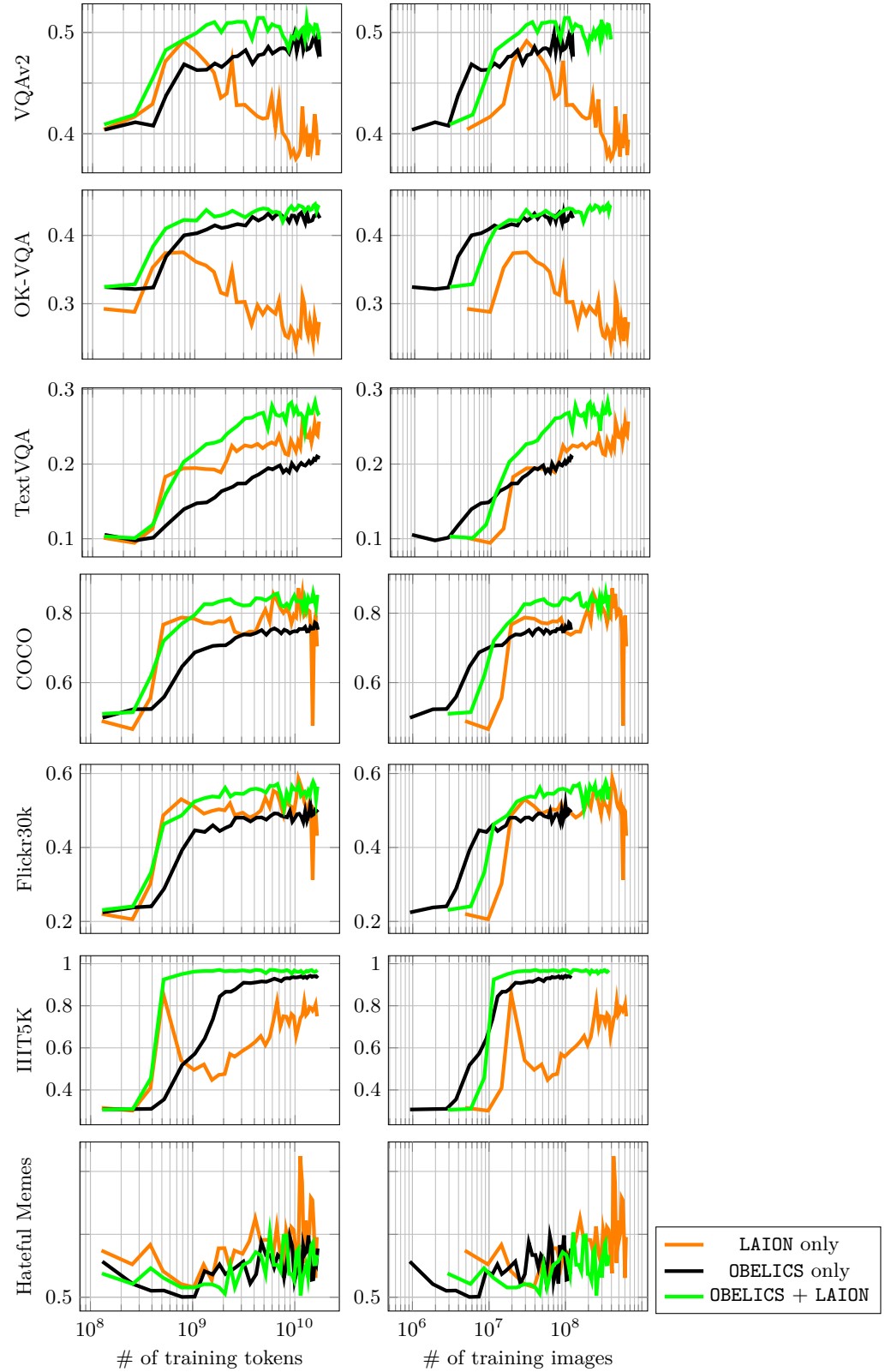

Figure 11: 4-shot performance through the training using `LAION` only, `OBELICS` only and a mixture of both. The training sequences from multimodal documents and the packed sequences obtained from image-text pairs have different numbers of images but the same number of tokens. Thus, we plot the performance over two log x-axes.

### A.5 License and Author Statement

We release the dataset under a CC-BY license and Terms of Use that require disclosure of when the dataset is used for the purpose of training models. This license is not intended to replace the licenses of the source content, and any use of content included in the dataset must comply with the original licenses and applicable rights of its data subjects.

The purpose of this statement is to clarify the responsibilities and liabilities associated with the use of this dataset. While we have made every effort to ensure the accuracy and legality of the data contained within this dataset, we cannot guarantee its absolute completeness or correctness.

Therefore, if any rights, legal or otherwise, are violated through this dataset, including but not limited to copyright infringement, privacy violations, or misuse of sensitive information, we, the authors, assume no liability for such violations.

By utilizing this dataset, you agree that any consequences, legal or otherwise, arising from using this dataset will be the user's sole responsibility. You acknowledge that you will exercise due diligence and adhere to all applicable laws, regulations, and ethical guidelines when using the dataset.

By accessing, downloading, or using this dataset, you signify your acceptance of this statement and your commitment to abide by the terms and conditions of the CC-BY license.

If you disagree with the terms of this statement or the CC-BY license, you are not authorized to use this dataset.

The dataset will be hosted and maintained on the Hugging Face Hub.