# OpenReview forum: "OBELICS: An Open Web-Scale Filtered Dataset of Interleaved Image-Text Documents"
_NeurIPS.cc/2023/Track/Datasets_and_Benchmarks — NeurIPS 2023 Datasets and Benchmarks Poster_

### Official Review · Reviewer_jNjy · 2023-07-20
**OBELICS: An Open Web-Scale Filtered Dataset of Interleaved Image-Text Documents**

**Rating:** 6
**Confidence:** 4
**Correctness:** None
**Clarity:** The paper is well written.

**Strengths:**

1. The proposed large-scale image-text is useful for the community.
2. This paper comprehensively analyzes the proposed datasets.
3. This paper is well-written and easy to understand the process of collecting this dataset.


**Additional Feedback:**

See “Opportunities For Improvement”.

**Documentation:**

The description of the dataset is comprehensive.

**Opportunities For Improvement:**

1. There is no indication of datasets licenses; hence if anybody wants to use this dataset, we need to check the license itself.
2. Lack of comparison with more similar image-text datasets. Specifically, the authors should train a baseline model with different image-text datasets and compare their performance on different applications.
3. It will be better if the authors provide several downstream applications.


**Relation To Prior Work:**

The related work section has discussed the related tasks. Some comparisons are also included in this paper.

**Summary And Contributions:**

The authors propose OBELISC, which is an innovative dataset and aims at supporting open-source large multimodal models. It comprises a vast collection of filtered interleaved multimodal web documents. To showcase the potential of models trained on multimodal documents, the authors trained a model that proves to be a compelling alternative to closed datasets. The availability of open datasets containing multimodal documents from diverse sources, with an emphasis on scale, quality, and diversity, plays a pivotal role in fostering the development of competitive open models.

---

> ### Author Response · Authors · 2023-08-19
> **Answer to Reviewer jNjy**
>
> Thank you reviewer jNjy for your comments.
>
> We will go through them one by one.
>
> > There is no indication of datasets licenses; hence if anybody wants to use this dataset, we need to check the license itself.
>
> 1 - We might have uploaded by mistake a wrong version of the supplementary materials, sorry about that.
> In the current version, there is a section at the very end about licensing and hosting. The license is also mentioned directly on the dataset page at https://huggingface.co/datasets/HuggingFaceM4/OBELICS.
>
> > Lack of comparison with more similar image-text datasets. Specifically, the authors should train a baseline model with different image-text datasets and compare their performance on different applications.
>
> 2 - You are correct that we only compared OBELICS to mmc4. However, it is the only open dataset that came close to our scale, and therefore the only dataset that we can sensibly compare OBELICS with. Interleaved image-text datasets are a relatively new development in multimodal machine learning and we are excited to provide OBELICS as a research artifact to the research community.
>
>
> > It will be better if the authors provide several downstream applications.
>
> 3 - Could you rephrase your question? If you are referring to the applications of the dataset, we can give several examples.
> First, as we showed in the paper, it allowed to boost the performance of models trained on it on reasoning tasks.
> Second, if one wants to train a vision language model from scratch, or at least not start from a pre-trained language model, we believe that our dataset is a strong starting point because it contains long texts and documents that resemble the ones from Wikipedia (see Figure 5 where we plotted the perplexity distributions).
> Third, we can imagine a scenario where it can be useful for information retrieval. For example, given an image, we could find documents containing an image close to it, and extract the text from them to obtain information about the picture (a historical figure in it, for example).
>
> Now, if you are referring to the applications of a model trained on this dataset, we can think about a multimodal chatbot assistant:
> - give travel tips given an image where we want to go to
> -give the recipe given a picture of a dish
> -write a text meme
> -answer questions about images
> -...
> We are going to release a large model trained on OBELICS and hope these points will be clearer.

---

### Official Review · Reviewer_m4Ux · 2023-07-20
**Open source multimodal documents dataset with detail description**

**Rating:** 7
**Confidence:** 4
**Correctness:** None
**Clarity:** Yes

**Strengths:**

1. A very detailed explanation for creating OBELISC. The authors also provide the reproduction code for OBELISC. I like them since they make readers/users deeply grasp the characteristics of the proposed dataset.

2. Interesting dataset analysis. I didn't know some documents have a lot of images but a small portion of text, like Figure 10. The topic modeling part and qualitative assessment are also very interesting to me.

3. Comparison with a concurrent work mmc4.

**Additional Feedback:**

None

**Documentation:**

Need to include "Datasheet" or "intended uses, hosting, licensing, and maintenance plan" in the supplementary materials.

**Opportunities For Improvement:**

Some major comments:
1. From Table 1, mmc4 seems to be the major competitor or alternative of OBELISC. However, there are only a few performance comparisons between them (L295-301). I wonder whether OBELISC is better than mmc4 or competitive with mmc4. I would like to check more performance comparisons. I know performance comparisons would be difficult since mmc4 is a concurrent work.

2. Flamingo showed results on much more tasks/datasets (16 tasks), but this work only showed results on 6 datasets in Table 2. I wonder if OBELISC is also effective for other tasks/datasets.

3. For most cases in Table 2, Flamingo is better than "Ours". The performance gap is almost 10% in HatefulMemes, and it seems not a "minor performance lag".


Some minor comments:
1. These sentences seem to need citations (L20) "Models trained on these web documents outperform vision and language models trained solely on image-text pairs on various benchmarks. They can also generate long and coherent text about a set of multiple images."

2. From L130 "We obtain 3.6 billion image links and successfully download 55% of them", I wonder why the authors downloaded only 55%. Is this sentence means that the authors failed to download the remaining 45% for some reason? I wonder about the specific reason.

3. From L171 "common advertisements encountered during the crawling process", some images are from common advertisements. Are they not relevant to its text document? From L172 "To address this issue, we remove all images that appear more than ten times across the entire dataset", this indicates that 10 irrelevant images per common advertisement can be included as noisy data. If these irrelevant images can be excluded, this work may improve.

4. From Figure 1, at first glance, I misunderstood 1 image for 1 text paragraph (1 image-1 text paragraph pair), but it is not.
Placing text above the first image, like Figures 7/8/9, may prevent this misunderstanding.

5. Difficult to understand the y-axis label of Figure 3.

6. Change the order of Figures 4 and 5?

7. Figure 4 is the results on the subsets of datasets (100K documents). From Table 4, OBELISC has 155K documents from Wikipedia. These indicate sampling way for each dataset may affect the results of Figure 4.

**Relation To Prior Work:**

Yes

**Summary And Contributions:**

The main contribution of this work is to provide open-source large-scale interleaved Image-text documents (multimodal documents) named OBELISC (renamed from OBELI"CS" to OBELI"SC").
As far as I know, there is no public dataset for multimodal documents except concurrent work mmc4.
The authors describe the dataset construction process in detail and give some useful analysis.
This work also shows the potential of multimodal documents by comparing OBELISC to image-text pair datasets LAION.

---

> ### Author Response · Authors · 2023-08-19
> **Answer to Reviewer m4Ux (1/2)**
>
> Thank you Reviewer m4Ux for your interesting remarks and your interest!
>
> We start with your major comments.
>
> > From Table 1, mmc4 seems to be the major competitor or alternative of OBELISC. However, there are only a few performance comparisons between them (L295-301). I wonder whether OBELISC is better than mmc4 or competitive with mmc4. I would like to check more performance comparisons. I know performance comparisons would be difficult since mmc4 is a concurrent work.
>
> 1 - It would indeed be very interesting to train the same model on OBELICS vs mmc4.
> Since they are different, for example in terms of number of tokens, we might see results like a combination of both is the best, as we saw when we combined OBELICS and LAION.
> However, this is substantial work. We would first need to download ourselves the hundreds of millions of images, integrate the dataset in our format and codebase, and train a model on it, which is long and requires a lot of compute, even at the 7B scale.
> We tried our best to compare models trained on OBELICS vs mmc4 even though we were not expected to do so as concurrent work. Given the time constraints, we leave that experiment for future works,
>
> > Flamingo showed results on much more tasks/datasets (16 tasks), but this work only showed results on 6 datasets in Table 2. I wonder if OBELISC is also effective for other tasks/datasets.
>
> 2 - It is true that Flamingo evaluates on 16 tasks. 8 of them are video tasks, and we haven’t trained our modelon video datasets As such, we did not evaluate our model on video benchmarks.
> In the updated version of the paper, we report numbers in 0/4/8/16/32 shots for all the 8 vision/language benchmarks Flamingo is evaluated on, using the same reported splits and same evaluation setup.
>
> > For most cases in Table 2, Flamingo is better than "Ours". The performance gap is almost 10% in HatefulMemes, and it seems not a "minor performance lag".
>
> 3 - After properly finishing the training and doing the checkpoint selection, we updated the numbers in the paper and named the model IDEFICS.
> In 32-shot, with our largest model, we either perform better or obtain the same result as Flamingo on 4/8 tasks.
> It is true that we are far behind on Hateful Memes. It is however worth mentioning that we consistently obtained between 10 and 15 additional points on the validation set of the benchmark, and we do not know why there was such a big drop in performance when evaluating on the test set. It could be an error on our side or a coincidence.

---

> > ### Author Response · Authors · 2023-08-19
> > **Answer to Reviewer m4Ux (2/2)**
> >
> > Now we can move on to the minor comments:
> >
> > > These sentences seem to need citations (L20) "Models trained on these web documents outperform vision and language models trained solely on image-text pairs on various benchmarks. They can also generate long and coherent text about a set of multiple images."
> >
> > 1 - Thanks for pointing this out, in Flamingo, authors did an ablation where they showed that removing their interleaved image-text documents from their data mixture decreased the performance. We modified the paper to cite them again at this line.
> >
> > > From L130 "We obtain 3.6 billion image links and successfully download 55% of them", I wonder why the authors downloaded only 55%. Is this sentence means that the authors failed to download the remaining 45% for some reason? I wonder about the specific reason.
> >
> > 2 - You are correct that we failed to download 45% of the images.
> > We started from the HTML DOM trees to extract the links to the images we wanted to download.
> > These HTML files were archived by the Common Crawl (snapshots between 2019 and 2023), and a large number of websites have potentially disappeared (or the images have been removed, or the URL changed) by the time we tried to download them.
> > After trying to download all the images once, we tried again on all the failing ones, but this time we had a really low success rate of about 4%, meaning that it wasn’t worth it doing this operation another time, and that we simply couldn’t download a large part of the images.
> >
> > > From L171 "common advertisements encountered during the crawling process", some images are from common advertisements. Are they not relevant to its text document? From L172 "To address this issue, we remove all images that appear more than ten times across the entire dataset", this indicates that 10 irrelevant images per common advertisement can be included as noisy data. If these irrelevant images can be excluded, this work may improve.
> >
> > 3 - This is a good remark.
> > From what we have seen, the distribution of the number of times an image appears in the whole raw dataset is extremely unequal (think about it as the 60 richest persons in the world have as much money as the 3.5B poorest ones).
> > We noticed in our dataset that a few images were duplicated hundreds of thousands of times, and weighed a significant proportion of the whole set of images.
> > This is why we introduced a cutoff of 10 repetitions. This allowed us to remove all the images repeated multiple times that were completely unrelated to the articles, while keeping a good portion of the images repeated a few times, but that were actually related to the articles.
> >
> >
> > > From Figure 1, at first glance, I misunderstood 1 image for 1 text paragraph (1 image-1 text paragraph pair), but it is not. Placing text above the first image, like Figures 7/8/9, may prevent this misunderstanding.
> > 4 - We are not 100% sure of what you meant. Could you rephrase your question?. If it’s related to the block “Multimodal document”, the first image is placed first because it follows the order of the original web page. It also looks more like a logo and made sense to be at the top rather than in between two text paragraphs. But visually maybe you are saying that it would be better if it was in the order: paragraph_1, image_1, paragraph_2, paragraph_3, image_2?
> >
> > > Difficult to understand the y-axis label of Figure 3.
> >
> > 5 - It is indeed a bit hard to understand, and even more to formulate it concisely. Say if you consider 20 for the axis number (the maximum number of images you allow in a document). Then, we see that for mmc4, 25% (100% minus 75% in the y-axis) of the total images are present in documents with more than 20 images. Even if the number of documents containing more than 20 images is really small. This echoes what was said before about how a small number of documents / unique images can account for a significant proportion of the total number of images. We tried to mitigate this in our dataset.
> >
> > > Change the order of Figures 4 and 5?
> >
> > 6 - We changed the order of the figures.
> >
> > > Figure 4 is the results on the subsets of datasets (100K documents). From Table 4, OBELISC has 155K documents from Wikipedia. These indicate sampling way for each dataset may affect the results of Figure 4.
> >
> > 7 - Yes, this is a good remark. The 100k documents are taken at random, so the chances to take one from Wikipedia out of the 141M documents stay small (1/1000) and couldn’t change the distribution enough to modify the conclusions.
> >
> > Concerning the hosting and maintenance plan, it might be due to a mistake from us in the upload of the supplementary materials because there is, at least now in this new version of the paper, a section about this at the very end of the document. We also updated the dataset card at https://huggingface.co/datasets/HuggingFaceM4/OBELICS by adding some more details.
> >
> > We hope we've cleared things up!

---

> > > ### Comment · Reviewer_m4Ux · 2023-08-24
> > > **Thank you for your excellent answers to my comments!**
> > >
> > > I am glad that most of my concerns are addressed, and thus, I raise my rating.
> > > And never mind about the 4th minor comment. It was just my misunderstanding (I thought 1 paragraph per 1 image at first glance).

---

### Official Review · Reviewer_pbuU · 2023-07-22
**This paper introduces the OBELICS dataset, which is a large-scale filtered dataset of interleaved image-text documents that can be used to train multimodal models.**

**Rating:** 6
**Confidence:** 5
**Correctness:** Yes
**Clarity:** Yes

**Strengths:**

● This paper gives  a clear description of the data collection process and and provided an inspiration for cleaning crawled data from the web.


**Additional Feedback:**

See above weaknesses.

**Documentation:**

Yes

**Limitations:**

● There is no consistent abbreviation for paper [1]. It is called the Multimodal C4 dataset in Line 78, and mmc4 in Line 206 and Table 1.
● What is the difference between mmc4-ff and mmc4 in Table 1?

**Opportunities For Improvement:**

1. In lines 78-87, the authors note that the OBELICS dataset proposed in this paper is more recent than the Multimodal C4 dataset and includes a more comprehensive filter rule. However, the authors fail to explain the exact benefits of these two designs.
On one hand, I doubt whether the data in the Multimodal C4 dataset is outdated, as it started in 2019. If it is outdated, what specific aspects will it affect? On the other hand, will the filtering rules used in the Multimodal C4 dataset generate a large amount of noisy data, which will affect the pre-training of visual language models based on this dataset? The author did not provide quantitative experimental analysis on this issue. Therefore, I believe that the motivation is insufficient.
2. The dataset proposed in this article does not have many new features.


**Relation To Prior Work:**

What is the difference between mmc4-ff and mmc4 in Table 1?
[1] Zhu, W., J. Hessel, A. Awadalla, S. Y. Gadre, J. Dodge, A. Fang, Y. Yu, L. Schmidt, W. Y.
626 Wang, and Y. Choi (2023). Multimodal C4: An open, billion-scale corpus of images
627 interleaved with text. arXiv preprint arXiv:2304.06939.

**Summary And Contributions:**

This paper introduces the OBELICS dataset, which is a large-scale filtered dataset of interleaved image-text documents that can be used to train multimodal models.  The authors describe the creation and filtering process of the dataset, and compare the performance of models trained on OBELICS to those trained on other image-text datasets.

---

> ### Author Response · Authors · 2023-08-19
> **Answer to Reviewer pbuU**
>
> Thank you Reviewer pbuU for your questions and feedback!
>
> We would like to mention that Multimodal C4 is a concurrent work in accordance with the NeurIPS guidelines, and authors and not expected to compare against works that have been published within a span of 2 months.
>
> We argue that large-scale interleaved image-text datasets are going to play a big role in the development of large-scale multimodal models and see parallel works in that direct as a net benefit for the machine learning community.
>
> We will now try to answer your questions.
>
> > On one hand, I doubt whether the data in the Multimodal C4 dataset is outdated, as it started in 2019. If it is outdated, what specific aspects will it affect?
>
> You are right that Multimodal C4 is not “outdated”: we highlighted the decisions in constructing the datasets. OBELICS has a more recent content, and the knowledge of the COVID, Biden’s presidency, or anything that happened between 2019 and 2023. We also want to point out that, in practice, answering whether certain models trained on OBELICS vs Multimodal C4 have more or less factual knowledge of recent events is a very hard and nuanced question. Indeed, the multimodal models trained on these datasets are typically initialized from a language model that is trained on a different text corpora that can contain very recent documents.
>
> > On the other hand, will the filtering rules used in the Multimodal C4 dataset generate a large amount of noisy data, which will affect the pre-training of visual language models based on this dataset?
>
> Next, we will summarize the differences between Multimodal C4 and OBELICS.
> While our motivations are the same, the documents in the respective datasets end up being really different.
>
> In Multimodal C4, the texts are much shorter than in OBELICS. For one image, there are on average 73.5 text tokens, while 325.5 tokens for OBELICS (4.4 times more). These longer texts give more context, and resemble more the documents we find on the web.
>
> To build our dataset, we started directly from the HTML DOM tree, instead of the extracted texts for them. This allowed us to remove a significant part of the ads, spam, or unwanted content with the help of the HTML tags. That would be really hard to remove these afterward once the text is already extracted from the HTML file.
>
> As mentioned in the comment for Reviewer Jcmj, we found that not removing the ads or spam sentences does have a clear impact on the generations of the model we train on the dataset.
>
> We then designed many image and text filters that helped us to have documents of much better quality. We estimated the quality of the documents both qualitatively (with our visualization https://huggingface.co/spaces/HuggingFaceM4/obelics_visualization) and quantitatively, with the perplexity analysis of the text of OBELICS compared to some of the most used text datasets (The Pile, C4, OSCAR), as well as the text of MMC4.
> We see in Figure 5 that the distribution of perplexity scores of OBELISC documents aligns closely with the one from The Pile, which was designed to have high quality text through careful curation of the data sources, and are lower (meaning the documents resemble more the ones from Wikipedia) than the ones from C4, OSCAR, and Multimodal 4. On the contrary, we see in this figure that Multimodal C4 texts are the least similar to Wikipedia documents.
>
>
> Last, we applied a more careful image deduplication.
> We noticed that in MMC4 only 60% of the images are unique, and the rest is duplicated. We believe training on such a high number of duplicated images is detrimental for several reasons. First, we observed that duplicated images are much more likely spam or unrelated content to the document which was gathered during the crawling. Intuitively, training on these low quality images (with low alignment with the text) could hurt the model. Second, prior works cited in the paper showed that training on duplicated data allows reaching better performance faster.
>
> > There is no consistent abbreviation for paper [1]. It is called the Multimodal C4 dataset in Line 78, and mmc4 in Line 206 and Table 1. ● What is the difference between mmc4-ff and mmc4 in Table 1?
>
> We made it more clear that  “mmc4” refers to “Multimodal C4” Thanks for pointing this out.
> Mmc4-ff is a subset of mmc4 with fewer faces: the authors removed images of people’s face from the dataset.
>
> We also invite you to read the last part of our answer to Reviewer Jcmj, where we explain what we highlight the ethical considerations of building OBELICS.
>
> If you have any additional questions, we would be happy to answer them!

---

### Official Review · Reviewer_Jcmj · 2023-07-25
**OBELISC offers another compelling open-source multimodal dataset for image-text model training.**

**Rating:** 6
**Confidence:** 4

**Strengths:**

This paper is clear, well documented, and explained. I appreciate the citation-heavy writing of the authors and the robust nature of their filtering. Section 4 of the paper is also strong, and provides a great pointer to the supplementary material which contains a great depth of analysis.

- Another billion-scale multimodal dataset is always a win for the community at large.
- A strong section analysing the dataset continues to add to the literature focusing on understanding the data we train our models on.
- The results section fairly demonstrates that OBELISC provides value to the community and will likely be used.

**Additional Feedback:**

N/A

**Clarity:**

Paper is clear except for some confusing phrases and text. EG: 288-294

This paper is well written and fits with the standard of papers that should be at NeurIPS.

**Correctness:**

The dataset is constructed in a sound way, providing access to code also ensures validity.

**Documentation:**

Sufficient documentation for current standards. Providing access to code is fantastic.

A hosting and maintenance plan is not mentioned clearly, and insight into the future of this dataset, expansions or otherwise, would be useful.

**Ethics:**

Ethics is underrepresented in the paper. Large web-scale datasets are difficult to purge ethically dubious material from and this paper undergoes the sufficient steps. There is a lot of ethical improvement in our field to pursue, and it is always unfortunate to read a paper that treats ethical considerations as a checklist of items. I do not hold this against OBELISC but I would have prefered to see unique analyses of the ethical considerations required of a dataset described by longer-token-text-sequences.

**Limitations:**

The authors have sufficiently addressed limitations and negative impact of their work. However, expanding on the limitations of the dataset would be a key aspect for improvement.

**Opportunities For Improvement:**

Expanding the set of experiments performed in both analysing and evaluating the dataset for performance would be very useful. In particular:

1. Ablations for training on subsets or filters of the dataset would provide useful insights for future users.
2. Greater explanation of the unique filtering and ingestion process used to create the dataset would better establish this paper in the space.

**Relation To Prior Work:**

Paper follows other work well. Offers a novel featuring method, and some useful results for the community. Prior work is exceptionally well cited.

**Summary And Contributions:**

OBELISC follows in the footsteps of many an open-source multimodal image-text dataset. The most pivotal of the paper's contributions is a strong documentation of the filtering and screening methods used to filter text. Relying on the HTML DOM to extract high quality text data is an important focus of the paper and has seemingly resulted in decent performance.

While the paper is strong in many areas there are a lot of other experiments and ablations that would be incredible interesting to study. I believe this paper serves as a useful contribution to the field, but would like to set the norm of papers expanding the set of experiments and tests when introducing a new dataset. In particular, ablations around token-length-per-image, subsampling of OBELISC and alternatives, and other suggestions I make later in my review, would benefit this paper's position.

APOLOGIES -- I've accidentally submitted this review along with several others hence the TODO's -- they will be cleared ASAP (I am writing this review in a separate markdown document.)

---

> ### Author Response · Authors · 2023-08-19
> **Answer to Reviewer Jcmj (1/2)**
>
> Thank you Reviewer Jcmj for your constructive feedback.
>
> You are mentioning interesting experiments, particularly doing ablations like training on a chosen subset of the dataset, or testing the effects of the different filters on the resulting model we train.
>
> We have conducted early experiments studying the effects of filtering on the resulting trained model.
>
> First, we trained models on OBELICS without any filters applied and compared the model against a similar model trained on a filtered OBELICS dataset. We obtained significant performance improvements after filtering on various benchmarks. Additionally, we also observed improvements in the quality of the conditional generations. While the model trained on unfiltered data tend to generate sentences like “Posted on Facebook” or “Share on Twitter”, the model trained on filtered OBELICS did not show such a pronounced behavior. Studying generations from the model allowed us to iteratively add filters to capture flaws of the dataset and catch undesirable patterns in the web documents.
>
> Second, we also tried tuning the values of the thresholds for each filter in a data-driven fashion, by training the same models on data obtained from different sets of threshold values. That experiment turned out to be unsuccessful. We did not see statistically significant differences at the model scale we were using (3B scale) on any of the downstream benchmarks, nor in the model generations. Besides, deactivating one filter among all of them also had very little impact at this scale suggesting that studying each filter separately (as opposed to class of filters) will likely yield no actionable insights. We thus resorted to manually inspecting a set of documents to choose the threshold values.
>
> We hypothesize that we would see more differences at a bigger scale (more than 7B parameters), given more parameters would allow the model to capture more subtle differences in the dataset. Given the required training time and our summer availability, we unfortunately won’t be able to redo these experiments properly.

---

> > ### Author Response · Authors · 2023-08-19
> > **Answer to Reviewer Jcmj (2/2)**
> >
> > > Greater explanation of the unique filtering and ingestion process used to create the dataset would better establish this paper in the space.
> >
> > Do you have specific points and details we could clarify? We are not sure what “ingestion process” refer to.
> >
> >
> > > some confusing phrases and text. EG: 288-294
> >
> > In an updated version of the paper, we have rephrased the sentence you mentioned line 288-294.
> >
> > > A hosting and maintenance plan is not mentioned clearly, and insight into the future of this dataset, expansions or otherwise, would be useful.
> >
> > We plan on keeping hosting the dataset on the Hugging Face Hub and share updates to the dataset through the same method.
> >
> > > Ethics is underrepresented in the paper. Large web-scale datasets are difficult to purge ethically dubious material from and this paper undergoes the sufficient steps. There is a lot of ethical improvement in our field to pursue, and it is always unfortunate to read a paper that treats ethical considerations as a checklist of items. I do not hold this against OBELISC but I would have prefered to see unique analyses of the ethical considerations required of a dataset described by longer-token-text-sequences.
> >
> > We regret that our write-up gives the impression that ethical considerations are relegated to a checklist. That was not the intent and we have made this more clear the write-up.
> >
> > At the beginning of the project, we reflected on [ethical principles](https://huggingface.co/blog/ethical-charter-multimodal) guiding the project, including the creation of the dataset, in order to incorporate ethical values we agreed on. These values motivated the careful crafting of the content filters. For instance, we used the Spawning API to respect as much as possible the consent decisions of content creators or iterated significantly on filters around pornographic content.
> >
> > Exploring large-scale corpora is often a tedious process which contributes to the lack of transparency and lack of documentation around these artifacts. With that in mind, we built an [interactive visualization](https://atlas.nomic.ai/map/f2fba2aa-3647-4f49-a0f3-9347daeee499/ee4a84bd-f125-4bcc-a683-1b4e231cb10f) of OBELICS which allows browsing through a subset (11M documents) of the dataset and navigate the different topics covered. Yet, we note that despite our efforts, OBELICS contains a small proportion of documents that are not suitable for all audiences. For instance, one might find the cluster named “Sex” which predominantly contains descriptions of pornographic movies along with pornographic images. Other clusters would contain advertising for sex workers, or reports of violent shootings. In our experience, these documents represent a small proportion of all the documents.
> >
> > Due to the nature of our dataset (multimodal documents extracted from the web), OBELICS inherits the same ethical concerns of unlabeled text corpora crawled from the web: difficulty to document/inspect, presence of unintended biases, under-representation of certain demographics, etc. These concerns have been well documented for text corpora (https://arxiv.org/abs/2011.02832, https://dl.acm.org/doi/10.1145/3442188.3445922). Data audits have shed light on the some limitations and unintended biases contained in these text corpora (https://arxiv.org/abs/2010.14571, https://arxiv.org/abs/2104.08758). The augmentation of text corpora with interleaved images is a recent development of multimodal machine learning. We hope that our dataset along with exploration tools will serve as a solid ground for endeavors such as data audits. Existing works auditing large-scale multimodal datasets have focused on image-text pairs datasets (https://arxiv.org/abs/2110.01963) and highlight how curation and filtering decisions lead to biases (including racism and misogyny) in the resulting pairs. We believe that interleaved image-text datasets will play a significant role in the development of increasingly more capable multimodal models, and having large-scale versions of these datasets that are transparent, maintained and in open-access is critical.
> >
> > We also have evaluated the trained models as part of a red-teaming effort and a systematic evaluation of the generations produced by the model compared across the axis of gender and race. More specifically, the model was separately prompted to write a resume, a dating profile, and a headline about a person’s recent arrest based on their appearance. We studied the generations and analyzed the trends for each protected characteristic using [FairFace](https://arxiv.org/abs/1908.04913) and [StableBias](https://arxiv.org/abs/2303.11408). The details of these evaluations and insights will be made public as part of the model release. As an example, the model trained on OBELICS associates men more frequently than women with terms like “financial”, “development”, “product”, and “software”.

---

> > ### Comment · Reviewer_Jcmj · 2023-08-30
> >
> > Nit: Could you reference (or point out to me in case I missed it) a location where you show that you have run experiments for training on filters etc. If there is no disclosed location of this it might be a nice addition but not severely important. In general I support research releasing more experiments (even if the results were not significant) than fewer.
> >
> > As for tuning the values of the thresholds in a data-driven manner, it is nice to hear that this was experimented with. Compute and personal time considerations are absolutely important. Thank you for this update.

---

### Author Response · Authors · 2023-06-13
**Dataset Renaming**

Regrettably, due to constraints imposed by our legal team, we cannot utilize the name OBELICS for our dataset. As a result, we have decided to rename it as OBELISC.
To have functional associated links (including the GitHub repository, visualization, and dataset page), we kindly ask the reviewers to change "obelics" to "obelisc" in our submitted paper.
We apologize for any inconvenience caused by this change.

---

### Author Response · Authors · 2023-08-19
**Enhancements introduced in response to the initial submission**

In addition to the specific responses provided to each reviewer, we would like to offer a general overview of the modifications and enhancements introduced in response to the initial submission.

Firstly, we are pleased to announce that we have obtained authorization to officially name the dataset OBELICS, thereby replacing the previous term OBELISC.

Secondly, a significant addition is the incorporation of an interactive visualization tool dedicated to OBELICS. This tool facilitates exploration within a subset of the dataset, encompassing 11 million documents, enabling users to navigate through various covered topics.

Moreover, substantial attention was dedicated to both the models and their evaluations. Specifically, we did a checkpoint selection for the 80 billion parameter model, in addition to training a new 9 billion parameter model. Both models were compared on more benchmarks and in more configurations to Flamingo and OpenFlamingo v2. Last, we have conducted a bias analysis on the trained models.

---

### Decision · Program_Chairs · 2023-09-22

**Decision:**

Accept (Poster)

**Comment:**

This paper introduces the OBELICS dataset, a vast collection of 141 million web pages with interleaved image-text content extracted from Common Crawl. It also discusses the creation process, filtering rules, and its use in training large multimodal models, IDEFICS-9B and IDEFICS, with 9 and 80 billion parameters respectively, achieving competitive performance on multimodal benchmarks. All reviewers are in unanimous agreement that this paper meets the standards for acceptance in terms of its contribution to the dataset and its rigor. Therefore, I recommend accepting this paper.